# Parafibromin governs cell polarity and centrosome assembly in *Drosophila* neural stem cells

Qiannan Deng[1], Cheng Wang[1], Chwee Tat Koe[1¤], Jan Peter Heinen[2], Ye Sing Tan[1], Song Li[1], Cayetano Gonzalez[2,3], Wing-Kin Sung[4,5], Hongyan Wang [1,6,7]*

**1** Neuroscience & Behavioral Disorders Programme, Duke-NUS Medical School, Singapore, **2** Institute for Research in Biomedicine, The Barcelona Institute of Science and Technology, Barcelona, Spain, **3** Institució Catalana de Recerca i Estudis Avançats, ICREA, Barcelona, Spain, **4** Genome Institute of Singapore, Genome, Singapore, **5** Department of Computer Science, National University of Singapore, Singapore, **6** Dept. of Physiology, Yong Loo Lin School of Medicine, National University of Singapore, Singapore, **7** NUS Graduate School—Integrative Sciences and Engineering Programme (ISEP), National University of Singapore, Singapore

¤ Current address: Genome Institute of Singapore, Genome, Singapore
* hongyan.wang@duke-nus.edu.sg

**Data Availability Statement:** All relevant data are within the paper and its Supporting Information files.

## Abstract

Neural stem cells (NSCs) divide asymmetrically to balance their self-renewal and differentiation, an imbalance in which can lead to NSC overgrowth and tumor formation. The functions of Parafibromin, a conserved tumor suppressor, in the nervous system are not established. Here, we demonstrate that *Drosophila* Parafibromin/Hyrax (Hyx) inhibits ectopic NSC formation by governing cell polarity. Hyx is essential for the asymmetric distribution and/or maintenance of polarity proteins. *hyx* depletion results in the symmetric division of NSCs, leading to the formation of supernumerary NSCs in the larval brain. Importantly, we show that human Parafibromin rescues the ectopic NSC phenotype in *Drosophila hyx* mutant brains. We have also discovered that Hyx is required for the proper formation of interphase microtubule-organizing center and mitotic spindles in NSCs. Moreover, Hyx is required for the proper localization of 2 key centrosomal proteins, Polo and AurA, and the microtubule-binding proteins Msps and D-TACC in dividing NSCs. Furthermore, Hyx directly regulates the *polo* and *aurA* expression in vitro. Finally, overexpression of *polo* and *aurA* could significantly suppress ectopic NSC formation and NSC polarity defects caused by *hyx* depletion. Our data support a model in which Hyx promotes the expression of *polo* and *aurA* in NSCs and, in turn, regulates cell polarity and centrosome/microtubule assembly. This new paradigm may be relevant to future studies on Parafibromin/HRPT2-associated cancers.

## Introduction

The asymmetric division of stem cells is a fundamental strategy for balancing self-renewal and differentiation in diverse organisms including humans. The *Drosophila* neural stem cells (NSCs), also known as neuroblasts, have emerged as an excellent model for the study of stem

**Funding:** This work is supported by Singapore Ministry of Education Tier 2 MOE2018-T2-2-047 to H.W. and Spanish Ministerio de Ciencia, Innovacion y Universidades (PGC2018-097372-B-100) to C.G. The funders had no role in study design, data collection and analysis, decision to publish, or preparation of the manuscript.

**Competing interests:** The authors have declared that no competing interests exist.

**Abbreviations:** AEL, after egg laying; ALH, after larval hatching; Ana2, Anastral spindle 2; APF, after puparium formation; aPKC, atypical PKC; Arl2, ADP ribosylation factor like-2; Asl, Asterless; atms, Antimeros; Atu, Another transcription unit; AurA, Aurora-A; Baz, Bazooka; BDSC, Bloomington Drosophila Stock Center; Cdc73, Cell division cycle 73; ChIP-qPCR, chromatin immunoprecipitation coupled with quantitative PCR; Cnn, centrosomin; CNS, central nervous system; Ctr9, Cln three requiring 9; DGRC, Drosophila Genomics Resource Center; dsRNA, double-stranded RNA; EMS, ethyl methanesulfonate; GMC, ganglion mother cell; HRPT2, hyperparathyroidism type 2; Hyx, Hyrax; hyx-FL, full-length hyx; INP, intermediate neural progenitor; Insc, Inscuteable; MARCM, mosaic analysis with a repressible cell marker; Mira, Miranda; mRNA, messenger RNA; Msps, Mini spindles; MTOC, microtubule-organizing center; Mts, microtubule star; NSC, neural stem cell; PCM, pericentriolar material; Pins, Partner of inscuteable; Pon, Partner of Numb; PP2A, phosphatase 2A; Pros, Prospero; RNAi, RNA interference; RT-qPCR, reverse transcription quantitative real-time PCR; Sas-4, Spindle assembly abnormal 4; SDC-SIM, Spinning Disc Confocal-Structured Illumination Microscopy; Stg, String; VDRC, Vienna Drosophila Resource Center; α-tub, α-tubulin; γ-tub, γ-tubulin; γ-TURC, γ-tubulin ring complex.

cell self-renewal and tumorigenesis [1–5]. During asymmetric division, each NSC generates a self-renewing NSC and a neural progenitor that can produce neurons and glial cells [2]. Cell polarity is established by the apically localized Par complex, including atypical PKC (aPKC), Bazooka (Baz, the Drosophila homologue of Par3), and Par6 [6–8], as well as the Rho GTPase Cdc42 [9]. This protein complex displaces the cell fate determinants Prospero (Pros), Numb and their adaptor proteins Miranda (Mira) and Partner of Numb (Pon) to the basal cortex [10–14]. Another protein complex, including Partner of inscuteable (Pins), heterotrimeric G protein subunit Gαi, and their regulators, which is linked to the Par proteins by Inscuteable (Insc), is recruited to the apical cortex during mitosis [15–21]. Upon division, apical proteins segregate exclusively into the larger NSC daughter cell to sustain self-renewal, and basal proteins segregate into the smaller progenitor daughter cell to promote neuronal differentiation [1,22]. Such asymmetric protein segregation is facilitated by the orientation of the mitotic spindle along the apicobasal axis [2,23–29]. A failure in asymmetric divisions during development may result in cell fate transformation, leading to the formation of ectopic NSCs or the development of brain tumors [24,26,27,30–36].

The dysregulation of a few cell cycle regulators, such as Aurora-A kinase (AurA), Polo kinase (Polo), and Serine/Threonine protein phosphatase 2A (PP2A) results in disruption to NSC asymmetry and microtubule functions, leading to NSC overgrowth and brain tumor formation [27,29,35,37–43]. Moreover, ADP ribosylation factor like-2 (Arl2), a major regulator of microtubule growth, localizes Mini spindles (Msps)/XMAP215/ch-TOG and Transforming acidic coiled-coil containing (D-TACC) to the centrosomes to regulate microtubule growth and the polarization of NSCs [44].

Human Parafibromin/Cell division cycle 73 (Cdc73)/hyperparathyroidism type 2 (HRPT2) is a tumor suppressor that is linked to several cancers, including parathyroid carcinomas and hyperparathyroidism–jaw tumor syndrome, head and neck squamous cell carcinomas, as well as breast, gastric, colorectal, and lung cancers [45–48]. Somatic mutations in parafibromin have been found in 67% to 100% of sporadic parathyroid carcinomas [45]. Parafibromin is part of a conserved polymerase-associated factor complex that primarily regulates transcriptional events and histone modification [49,50]. Hyrax (Hyx), Drosophila Parafibromin, is essential for embryonic and wing development and is known to positively regulate Wnt/Wingless signaling pathway in wing imaginal discs by directly interacting with β-catenin/Armadillo [51]. Human Parafibromin, but not yeast Cdc73, rescues defects in wing development and the embryonic lethality caused by hyx loss-of-function alleles [51], suggesting that Parafibromin functions during development are conserved across metazoans. Interestingly, Parafibromin is expressed in both mouse and human brains, including the cortex, basal ganglia, cerebellum, and the brainstem [52], suggesting that Parafibromin may play a role in central nervous system (CNS) functions. However, the specific functions of Parafibromin in the nervous system are not established. Here, we investigate the role of Parafibromin/Hyx in the asymmetric division of NSCs during Drosophila larval brain development.

## Results

### Loss of *hyx* results in NSC overgrowth in the larval central brain

In a clonal screen of a collection of chromosome 3R mutants induced by ethyl methanesulfonate (EMS) [53], we identified 2 new *hyrax* (*hyx*) alleles—*hyx*$^{HT622}$ and *hyx*$^{w12-46}$—which produce NSC overgrowth phenotype in the central brain of *Drosophila* larvae at 96 h after larval hatching (ALH) (Fig 1A and 1B). *hyx/CG11990* encodes a highly conserved, 531-amino acid protein that is homologous to mammalian Parafibromin/Cdc73. The *hyx*$^{HT622}$ allele contains a 74-bp deletion from nucleotides 728 (immediately after amino acid 242) to 801, which results

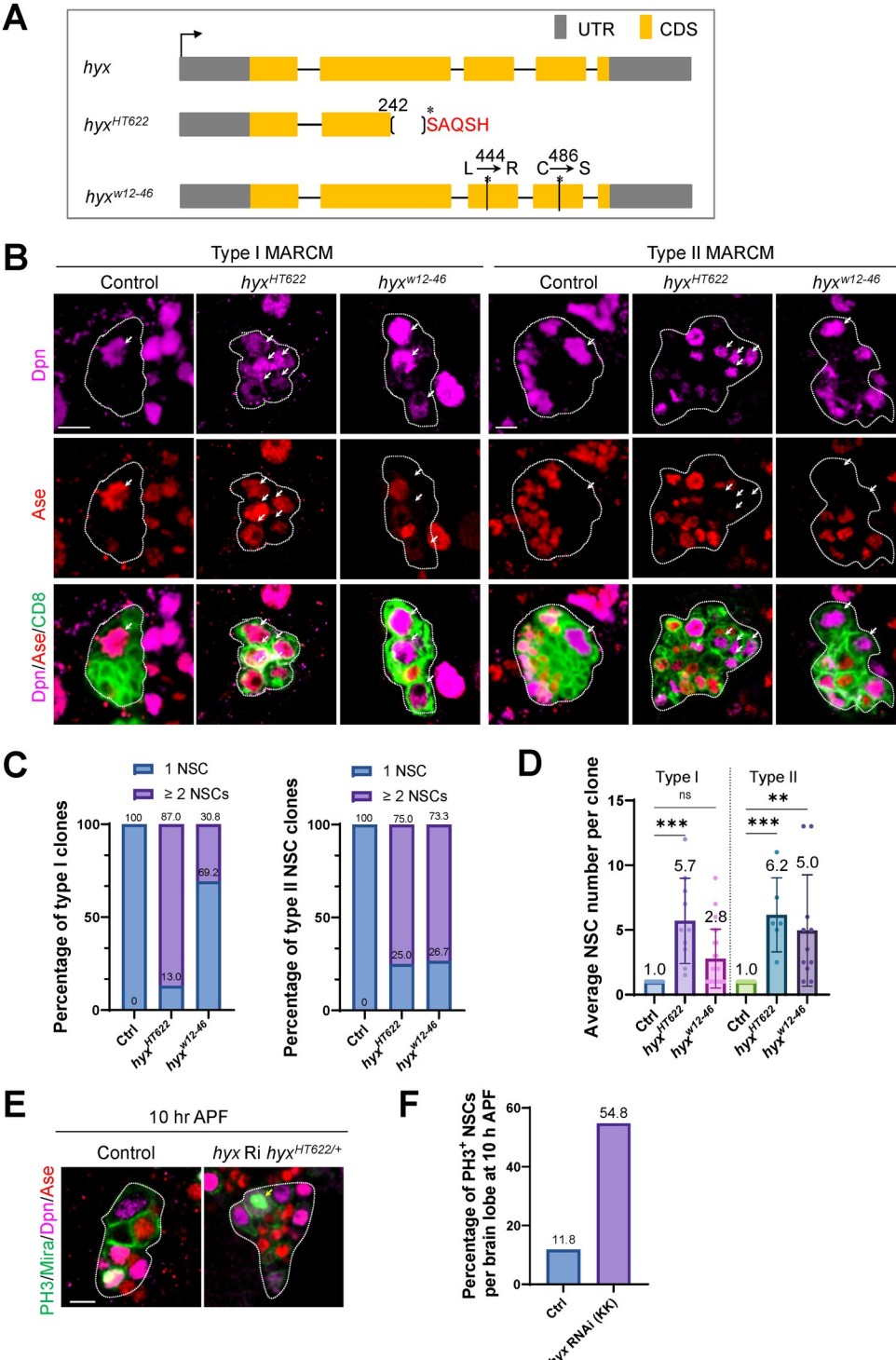

**Fig 1. Hyx regulates the homeostasis of central larval brain NSCs.** (**A**) An illustration of 2 *hyx* EMS alleles. (**B**) Type I and type II MARCM clones of various genotypes were labeled for Dpn, Ase, and CD8-GFP. Multiple NSCs were observed in *hyx*-depleted type I and type II MARCM clones. (**C**) Percentage of clones with ectopic NSCs for genotypes in B. Type I clones: control (FRT82B), *n* = 61; *hyx^HT622^*, 87%, *n* = 92; *hyx^w12-46^*, 30.8%, *n* = 68. Type II clones: control (FRT82B), *n* = 57; *hyx^HT622^*, 75%, *n* = 75; *hyx^w12-46^*, 73.3%, *n* = 66. (**D**) Average NSC number per clone (with SEM) for genotypes in B. Type I clones: control (FRT82B), 1, *n* = 61; *hyx^HT622^*, 5.7 ± 1.04, *n* = 92; *hyx^w12-46^*, 2.8 ± 0.47, *n* = 68. Type II clones: control (FRT82B), 1, *n* = 57; *hyx^HT622^*, 6.2 ± 1.17, *n* = 75; *hyx^w12-46^*, 5.0 ± 1.30, *n* = 66. (**E**) NSC lineages from control (*UAS-β-Gal* RNAi), and *hyx* RNAi *hyx^HT622/+^* pupal brains at 10 h APF under the control of *insc*-Gal4 driver were stained with PH3, Dpn, Ase, and Mira. PH3 and Mira were probed in the same channel. More NSCs positive for PH3 marker were observed in *hyx* RNAi *hyx^HT622/+^* papal brains (control, 11.8%, *n* = 34; *hyx* RNAi (KK) *hyx^HT622/+^*, 54.8%, *n* = 31). MARCM clones are outlined by dotted lines. Arrows indicate NSCs. Statistical significances

were determined by one-way ANOVA. In D type I, ***$p$ = 0.0005, ns = 0.2582; in D type II, ***$p$ = 0.0006, **$p$ = 0.0019. Scale bars: 5 μm. The underlying data for this figure can be found in the S1 Data. APF, after puparium formation; EMS, ethyl methanesulfonate; Hyx, Hyrax; MARCM, mosaic analysis with a repressible cell marker; Mira, Miranda; NSC, neural stem cell; RNAi, RNA interference.

in a frameshift mutation and, consequently, the generation of a stop codon at amino acid 248. This likely produces a truncated Hyx protein. The other *hyx* allele, *hyx^{w12-46}*, carries 2-point mutations at nucleotides 1331 (T to G) and 1456 (T to A), which causes amino acid substitutions—Leucine (L) to Arginine (R) at amino acid 444 and Cysteine (C) to Serine (S) at amino acid 486, respectively. As *hyx^{HT622}* and *hyx^{w12-46}* homozygotes are embryonically lethal, we generated MARCM (mosaic analysis with a repressible cell marker) clones [54] to examine the clonal phenotype at larval stages. Hyx protein in the clones was detected using guinea pig anti-Hyx antibodies that we generated against the N-terminal 1–176 amino acids of Hyx. In wild-type control clones, Hyx was predominantly localized to the nuclei of NSCs (S1A and S1B Fig; 0.74-fold in the nuclei, 0.26-fold in the cytoplasm, total intensity:1; *n* = 15 NSC) and their progeny, but not on mitotic structures (S1A and S1B Fig). In contrast, Hyx was undetectable in 91.3% and 40% of clones generated from *hyx^{HT622}* and *hyx^{w12-46}* alleles, respectively, and was dramatically reduced in the rest of the clones. The intensity of Hyx was significantly reduced to 0.15-fold in *hyx^{HT622}* NSCs (*n* = 25) and 0.32-fold in *hyx^{w12-46}* NSCs (*n* = 28), in contrast to 1-fold in control NSCs (S1A–S1C Fig; *n* = 20). Moreover, we performed western blot with protein extracts from FRT82B, *hyx^{HT622}*, and *hyx^{w12-46}* homozygous embryos, as *hyx* homozygous mutants do not survive to larval stages. Relative protein intensity of Hyx at 24 h after egg laying (AEL) was reduced to 0.31-fold and 0.26-fold in both *hyx^{HT622}* and *hyx^{w12-46}*, respectively (S1D and S1E Fig; control, 1-fold). Maternal Hyx might partially contribute to the detected Hyx proteins in these samples. Hyx levels were also significantly reduced in *hyx* RNAi (RNA interference) in *hyx^{HT622}*/+ background to 0.22-fold in third instar larval brains driven by NSC-specific driver *insc*-Gal4 (S1D and S1E Fig). This western blot result is consistent with our immunofluorescence data that Hyx is dramatically diminished with weak signal in these *hyx* mutants at the third instar stage (S1A–S1C Fig). Therefore, these observations indicate that *hyx^{HT622}* and *hyx^{w12-46}* are 2 strong loss-of-function alleles.

In the central brain of *Drosophila* larvae, there are at least 2 types of NSCs, both of which divide asymmetrically [55–57]. Each type I NSC generates another NSC and a ganglion mother cell (GMC) that gives rise to 2 neurons, while each type II NSC produces an NSC and a transient amplifying cell (also known as an intermediate neural progenitor or INP), which, in turn, go through a few cycles of asymmetric divisions to produce GMCs [55–58]. In the wild-type control, only 1 NSC is maintained in each of type I or type II MARCM clones (Fig 1B–1D). However, ectopic NSCs were observed in 87% of type I clones and 75% of type II clones generated from the *hyx^{HT622}* allele (Fig 1B). Similarly, supernumerary NSCs were observed in both type I and type II NSC lineages from *hyx^{w12-46}* clones (Fig 1B). Ectopic NSCs per clone were observed in type I *hyx^{HT622}* (5.7) and *hyx^{w12-46}* (2.8) clones and type II *hyx^{HT622}* (6.2) and *hyx^{w12-46}* (5.0) clones (Fig 1D). In addition, knockdown of *hyx* by 2 independent RNAi lines, under the control of an NSC driver *insc*-Gal4, led to the formation of multiple NSCs in both type I and type II lineages (S1F and S1G Fig). Moreover, the NSC overgrowth phenotype in *hyx^{HT622}* and *hyx^{w12-46}* mutants was fully rescued by the overexpression of a wild-type *hyx* transgene (S2A and S2B Fig). Therefore, our finding shows that *hyx* prevents NSC overgrowth in both type I and type II lineages.

Next, we wondered whether the supernumerary NSCs detected in *hyx*-depleted larval brains could persist in proliferation in pupal stages. At 10 h after puparium formation (APF),

88.2% of the NSCs in pupal brains were negative for PH3 (Fig 1E and 1F). In contrast, 54.8% of *hyx* RNAi *hyx*^HT622/+^ NSCs were proliferative and marked by PH3, suggesting that increased number of *hyx*-depleted NSCs were still actively dividing in early pupal stages. This phenotype might be due to an NSC decommissioning defect or abnormal NSCs that were generated by symmetric divisions acquired proliferative potential. Temporal transcription factors are known to schedule Pros-dependent cell cycle exit of NSCs at the end of larval stages [59]. The abnormal *hyx*-deficient NSCs often undergo symmetric division, which might result in a disruption of temporal factor transition and, in turn, continued proliferation after pupal formation.

Remarkably, the ectopic NSC phenotype observed in *hyx*^HT622^ larval brains was completely rescued by the overexpression of Parafibromin/HRPT2, the human counterpart of Hyx (S2C Fig). Likewise, the NSC overgrowth phenotype observed in *hyx* knockdown brains was fully restored by the introduction of human Parafibromin/HRPT2 in both type I and type II NSC lineages (S2D Fig). Therefore, Parafibromin/Hyx appears to have a conserved function in suppressing NSC overgrowth.

Parafibromin/Cdc73 (Hyrax/Hyx in *Drosophila*) is a component of the Paf1 complex, an evolutionarily conserved protein complex that functions in gene regulation and epigenetics [60–63]. The Paf1 complex also consists of other core subunits Paf1 (Antimeros/atms in *Drosophila*), Leo1 (Another transcription unit/Atu in *Drosophila*), Cln three requiring 9 (Ctr9 in *Drosophila*), and Rtf1 [64,65]. We sought to analyze the function of other components of the Paf1 complex in the larval central brains. Surprisingly, although Ctr9 is required to terminate the proliferation of *Drosophila* embryonic NSCs [33,66], no ectopic NSCs were observed in *ctr9*^12P023^ type I and type II MARCM clones (S2E Fig). Similarly, knockdown of *ctr9*, *atms*, or *atu* under the control of *insc*-Gal4 did not generate supernumerary NSCs in either type I or type II lineages in the larval central brains (*n* = 5 for all). Interestingly, knocking down *rtf* resulted in a weak ectopic NSC phenotype in type II lineages, without affecting type I NSC lineage development (*rtf1* RNAi/BDSC#34586: type II, 31.2%, **n** = 64 and *rtf1* RNAi/BDSC#34850: type II, 7%, *n* = 43). Therefore, our results suggest that Parafibromin/Hyx might prevent NSC overgrowth independent of Paf1 complex function during *Drosophila* brain development.

## Hyx is essential for the asymmetric division of NSCs

The generation of ectopic NSCs in the absence of Hyx function was not due to INP dedifferentiation, as no type II NSCs were generated in both control and *hyx* RNAi/V103555 derived INP clones (S3A Fig). *hyx* expression was efficiently down-regulated in these INP clones upon *hyx* knockdown (S3B Fig). Next, we assessed whether *hyx* is required for the asymmetric division of NSCs. In wild-type control metaphase NSCs, apical proteins such as aPKC, Insc, Baz/ Par3, Par6, and Pins were localized asymmetrically in the apical cortex (Fig 2A–2F). By contrast, aPKC in *hyx*^HT622^ and *hyx*^w12-46^ metaphase NSCs was completely delocalized from the apical cortex to the cytoplasm (Fig 2A and 2B). Similarly, other apical proteins including Insc, Baz, Par6, and Pins in *hyx*^HT622^ metaphase NSCs were no longer localized asymmetrically in the apical cortex, exhibiting weak or punctate signals in the cytoplasm (Fig 2C–2F).

Next, we examined the localization of basal proteins in *hyx* mutant NSCs. Mira was asymmetrically localized in the basal cortex in 100% of wild-type NSCs during metaphase, but its basal localization was severely disrupted in metaphase NSCs of *hyx*^HT622^ and *hyx*^w12-46^ clones (Fig 2A and 2B). Similarly, Numb and Brat lost their asymmetric basal localization and were observed in the cytoplasm in metaphase NSCs of *hyx*^HT622^ clones (Fig 2C and 2G).

Similarly, *hyx* knockdown by 2 independent RNAi lines disrupted NSC apicobasal polarity. aPKC, Par6, Baz, Insc, and Pins were delocalized from the apical cortex in all metaphase NSCs

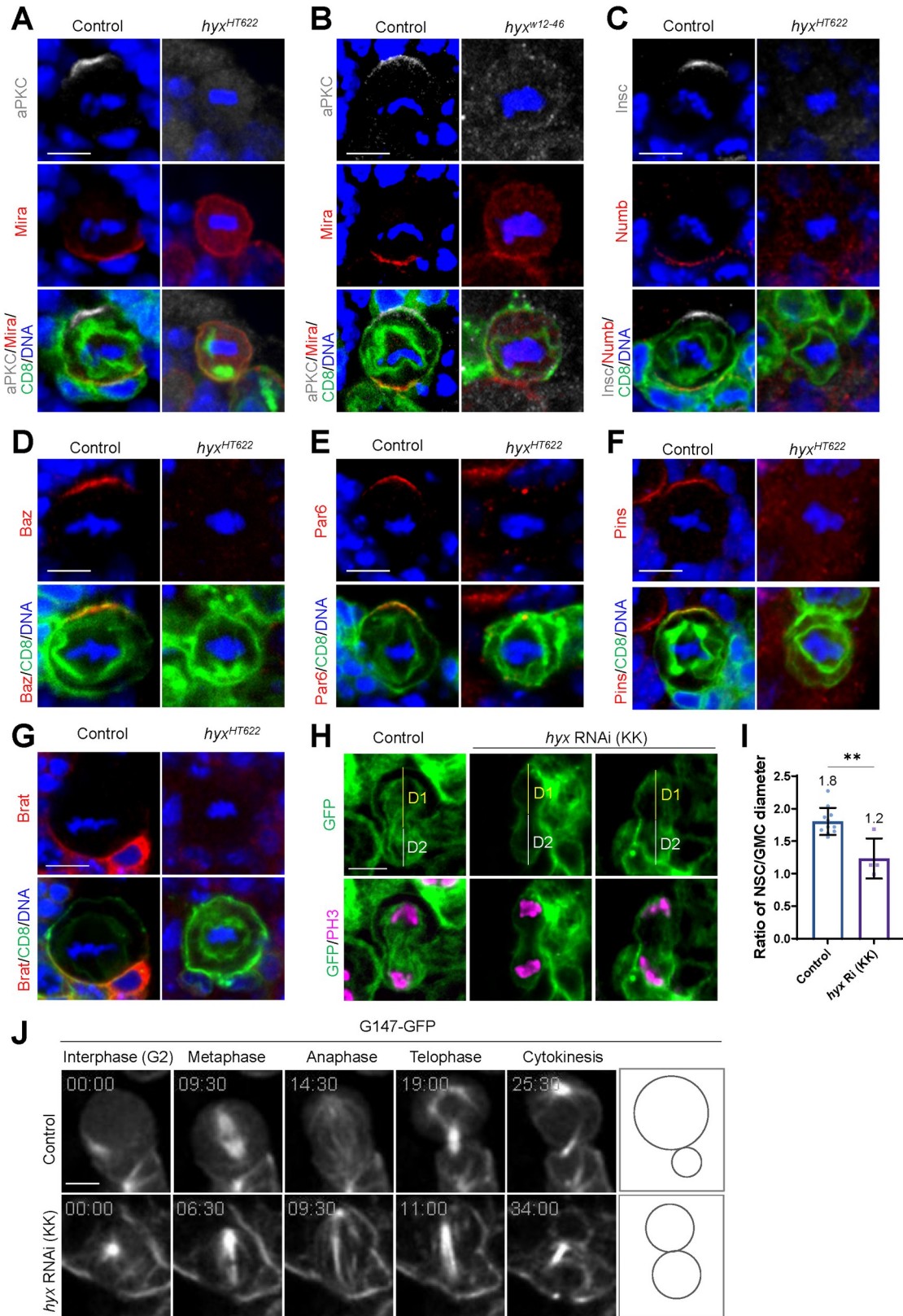

**Fig 2. Hyx is essential for asymmetric cell division of NSCs. (A)** Metaphase NSCs of control (FRT82B; $n$ = 50) and $hyx^{HT622}$ MARCM clones were labeled for aPKC, Mira, and DNA. $hyx^{HT622}$, aPKC delocalization, 100%; Mira delocalization, 81.6%; $n$ = 49

for both. (**B**) Metaphase NSCs of control (FRT82B; *n* = 45) and *hyx*$^{W12-46}$ MARCM clones were labeled for aPKC, Mira, and DNA. 100% delocalization of aPKC and Mira in *hyx*$^{W12-46}$, *n* = 50 for both. (**C**) Metaphase NSCs of control (FRT82B; *n* = 47) and *hyx*$^{HT622}$ MARCM clones were labeled for Insc, Numb, and DNA. 100% delocalization of Insc and Numb in *hyx*$^{HT622}$, *n* = 49 for both. (**D**) Control (FRT82B; *n* = 46) and *hyx*$^{HT622}$ metaphase NSCs were labeled for Baz and DNA. 100% Baz delocalization in *hyx*$^{HT622}$ (*n* = 49). (**E**) Control (FRT82B; *n* = 50) and *hyx*$^{HT622}$ metaphase NSCs were labeled for Par6 and DNA. 100% delocalization of Par6 in *hyx*$^{HT622}$ (*n* = 49). (**F**) Metaphase NSCs from control (FRT82B) and *hyx*$^{HT622}$ were labelled for Pins and DNA. 100% delocalization of Pins in *hyx*$^{HT622}$; *n* = 49 for both genotypes. Clones were marked by CD8-GFP. (**G**) Metaphase NSCs from control (FRT82B; *n* = 48) and *hyx*$^{HT622}$ were labelled for Brat and DNA. 100% Brat delocalization in *hyx*$^{HT622}$ (*n* = 50). (**H**) Telophase NSCs from control (*UAS-β-Gal* RNAi) and *hyx* RNAi (KK/V103555) under the control of *insc*-Gal4 driver were labeled for PH3 and CD8-GFP. D1 and D2 indicates NSC and GMC diameter, respectively. (**I**) Ratio of NSC to GMC diameter (with SD) for genotypes in H. Control: 1.8 ± 0.21 μm, *n* = 11 NSC; *hyx* RNAi, 1.2 ± 0.31 μm, *n* = 4 NSC. (**J**) Still images of time-lapse imaging of control (G147-GFP/+; *n* = 15 and S1 Movie) and *hyx* RNAi (KK/V103555; *n* = 5 and S2 Movie) NSCs expressing G147-GFP under the control of *insc*-Gal4 at 48 h ALH. G2 phase, metaphase, anaphase, telophase, and cytokinesis were shown. Statistical significances were determined by unpaired two-tailed Student *t* test. \*\**p* = 0.0011. Scale bar: 5 μm. The underlying data for this figure can be found in the S1 Data. ALH, after larval hatching; aPKC, atypical PKC; Baz, Bazooka; GMC, ganglion mother cell; Hyx, Hyrax; Insc, Inscuteable; MARCM, mosaic analysis with a repressible cell marker; Mira, Miranda; NSC, neural stem cell; Pins, Partner of inscuteable; RNAi, RNA interference.

of both *hyx* RNAi lines (S2C–S2I Fig). In addition, the localization of Mira, Numb, and Brat proteins at the basal cortex was also disrupted in metaphase NSCs upon *hyx* knockdown (S2C–S2E and S2I Fig). Taken together, our data indicate that Hyx orchestrates NSC polarity by regulating the proper localization of both apical and basal proteins.

To examine the daughter cell size asymmetry, we measured the ratio of NSC to GMC diameter at telophase. Cells were outlined by a membrane marker CD8-GFP driven by *insc*-Gal4. The NSC-to-GMC diameter ratio in *hyx* RNAi telophase NSCs was significantly reduced to 1.2 (Fig 2H and 2I; *n* = 4) compared with 1.8 (*n* = 11) in control telophase NSCs. These data indicate that *hyx* depletion disrupts daughter cell size asymmetry of the NSCs. Given that polarization of NSCs is an essential prerequisite for their asymmetric division, we sought to investigate whether *hyx* depletion could result in the symmetric division of NSCs. To this end, we took advantage of a microtubule-binding protein Jupiter-GFP (also known as G147-GFP), which is controlled under its endogenous promoter [67]. In live, whole-mount larval brains that expressed G147-GFP, control NSCs always divided asymmetrically to produce 2 daughter cells with distinct cell sizes (Fig 2J and S1 Movie). By contrast, all NSCs in *hyx* RNAi divided symmetrically to generate 2 daughter cells with similar cell sizes (Fig 2J and S2 Movie). These observations indicate that *hyx*-depleted NSCs divide symmetrically, leading to NSC overgrowth. However, the allograft transplantation of *hyx*$^{HT622}$ (*n* = 29 host) and *hyx*$^{w12-46}$ (*n* = 30 host) homozygous clones and *hyx* RNAi (*n* = 30 host) larval brain cells did not induce tumors. Perhaps *hyx*-depleted cells are small in size and with altered cell fate and have reduced cell growth, thus unable to expand in this tumor assay.

The protein levels of the phosphorylated Akt (Ser505) (P-Akt) were unaffected in *hyx* RNAi NSCs as compared with that in control NSCs (S4A and S4B Fig; 1.0-fold versus 1.02-fold). In addition, protein levels of a cell cycle regulator String (Stg)/Cdc25 in *hyx* RNAi NSCs were unaffected (S4C and S4D Fig; 1.01-fold versus 1.0-fold in control). Similarly, microtubule star (Mts), the PP2A catalytic subunit C, was unaffected in *hyx*$^{HT622}$ NSCs (S4E and S4F Fig; intensity normalized against Dpn: 0.89 in control versus 0.78 in *hyx*$^{HT622}$). These data suggest that Hyx does not seem to regulate PI3K/Akt pathway, Cdc25, or Mts in NSCs.

## Hyx maintains interphase microtubule asters and the mitotic spindle

When analyzing the asymmetric division, we noticed that *hyx*-depleted NSCs formed shorter mitotic spindles. The mean spindle length in control metaphase NSCs was 9.81 ± 0.94 μm, whereas in *hyx* RNAi (V103555) NSCs, the mean spindle length was dramatically shortened to 6.84 ± 1.46 μm (Fig 3A and 3B). Consistent with this observation, spindle lengths in metaphase

*hyx^HT622* and *hyx^w12-46* NSCs were much shorter than that in the control (Fig 3C and 3D). When normalized against the cell diameter of NSCs, the mitotic spindle length was still significantly shortened in *hyx*-depleted NSCs. The relative spindle length was significantly shortened to 0.74-fold in *hyx* RNAi NSCs (Fig 3E; *n* = 15 NSC) compared with 0.82-fold in control NSCs (*n* = 32 NSC). Similarly, the ratio of spindle length to cell diameter was significantly reduced to 0.70-fold in both *hyx^HT622* (*n* = 13 NSC) NSCs and *hyx^w12-46* NSCs (*n* = 21 NSC) when compared with 0.79-fold in control NSCs (Fig 3F; *n* = 22 NSC).

These observations prompted us to examine whether Hyx is important for microtubule assembly in NSCs. We sought to determine whether Hyx regulates the formation of microtubule asters in interphase NSCs. A wild-type interphase NSC forms 1 major microtubule aster marked by α-tubulin (α-tub). Asters are assembled by the microtubule-organizing center (MTOC), also known as centrosomes, of cycling NSCs labeled by a centriolar protein called Asterless (Asl; Fig 3G). Strikingly, the vast majority of interphase NSCs in *hyx^HT622* and *hyx^w12-46* clones either failed to organize a microtubule aster or formed weak microtubule asters (Fig 3G). The astral microtubule intensity marked by α-tub was dramatically reduced to 14.7 (a.u.) and 27.6 (a.u.) in *hyx^HT622* (*n* = 23 NSC) and *hyx^w12-46* NSCs (*n* = 23 NSC), respectively, significantly lower than 100.7 (a.u.) in control NSCs (Fig 3H; *n* = 7 NSC). Likewise, in *hyx* knockdown clones, 50% of NSCs failed to form microtubule asters and 40.9% of NSCs only assembled weak microtubule asters during interphase (S5A Fig). Overall, we show that Hyx is important for the formation of interphase microtubule asters.

The shortened mitotic spindles and defects in the assembly of microtubule asters upon *hyx* depletion suggested that Hyx might regulate microtubule growth in dividing NSCs. To this end, we performed a microtubule regrowth assay by "cold" treatment of larval brains on ice—for efficiently depolymerizing microtubules in NSCs—followed by their recovery at 25˚C to allow microtubule regrowth in the course of time. In both control and *hyx* RNAi interphase NSCs treated with ice (t = 0), no astral microtubules were observed and only weak residual microtubules labelled by α-tub remained at the centrosome (Fig 4A). The centrosomes in 76.3% of these *hyx* RNAi interphase NSCs were absent or much smaller in size, suggesting that the MTOC was compromised upon *hyx* depletion (Fig 4A). In control interphase NSCs, robust astral microtubules were observed around the centrosome, at various time points following recovery at 25˚C (Fig 4A). By contrast, the vast majority of *hyx* RNAi interphase NSCs reassembled scarce microtubule bundles without detectable MTOCs, even 120 s after recovery at 25˚C (Fig 4A).

Next, we examined microtubule regrowth in mitotic NSCs. Upon treatment with ice (t = 0), spindle microtubules were effectively depolymerized with residual microtubules marking the centrosomes/spindle poles, in all control and *hyx* RNAi metaphase NSCs (Fig 4B). Consistent with poor centrosome assembly during interphase, the centrosomes of 98.0% of *hyx* RNAi metaphase NSCs were deformed with irregular shapes (Fig 4B). In all control metaphase NSCs, intense spindle microtubules were reassembled around centrosomes and chromosome mass from as early as 30 s following recovery at 25˚C; the mitotic spindle completely reformed at 2 min following recovery (Fig 4B). In contrast, the majority of *hyx* RNAi metaphase NSCs assembled scarce spindle microtubule mass following recovery; at 2 min after recovery, only 14.6% of metaphase NSCs formed mitotic spindles, which were still shorter and thinner than the spindles formed in the control NSCs. MTOCs remained weak or absent in these *hyx*-depleted NSCs (Fig 4B). Quantification of microtubule intensity suggested that in all of the time points (except for t = 0), microtubule intensity in *hyx*-depleted NSCs at both interphase and metaphase were significantly reduced compared with the control (Fig 4B and 4D). Taken together, we propose that Hyx plays a central role in the formation of interphase microtubule asters and the mitotic spindle by promoting microtubule growth in NSCs.

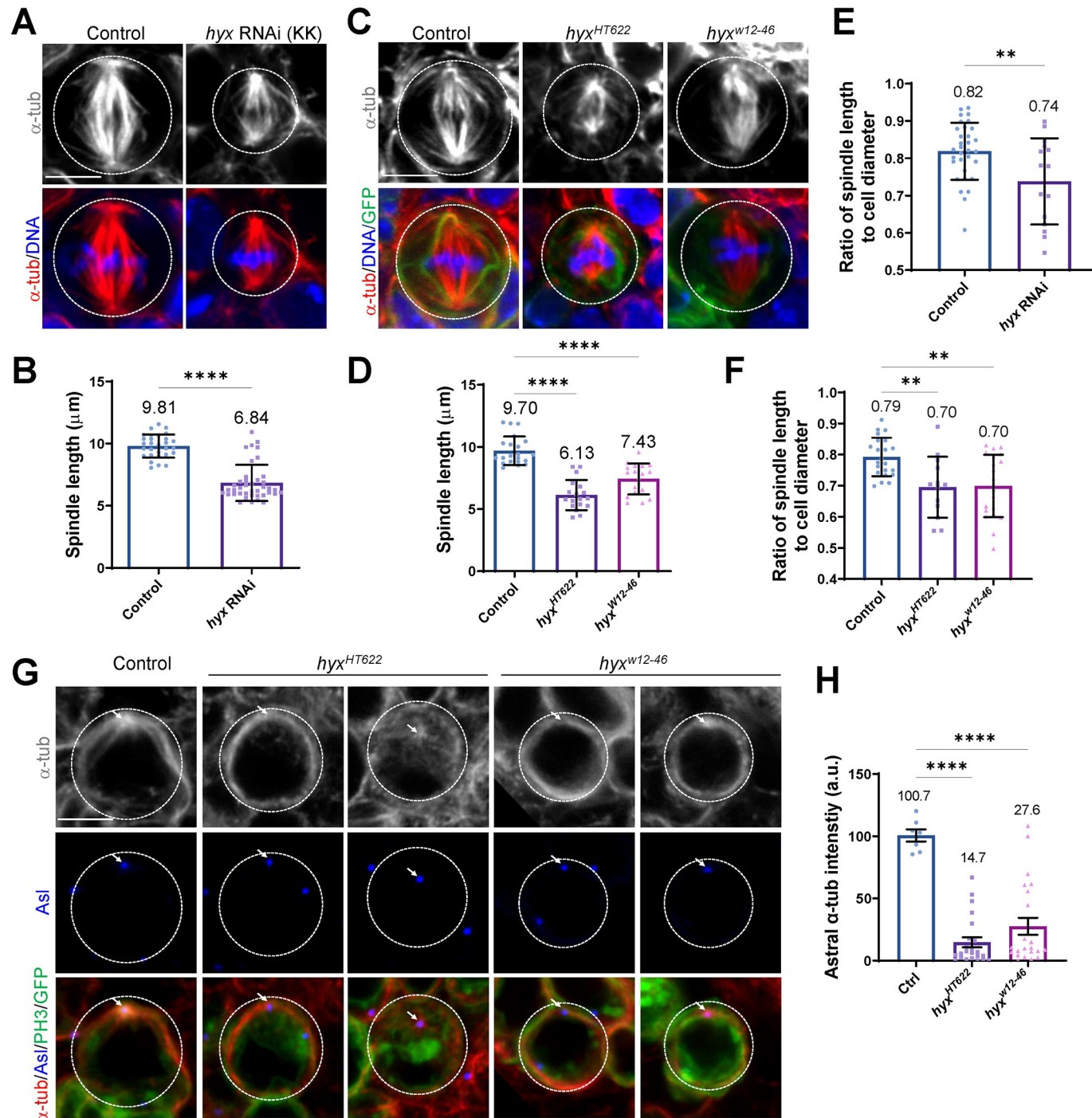

**Fig 3. Hyx is required for the formation of the microtubule aster and the mitotic spindle in NSCs.** (**A**) Metaphase NSCs of control (*UAS-β-Gal* RNAi) and *hyx* RNAi (KK/V103555) under the control of *insc*-Gal4 were labelled for α-tub and DNA. (**B**) Quantification of spindle length (with SD) observed in A. Spindle length: control, 9.81 ± 0.94 μm, *n* = 29; *hyx* RNAi, 6.84 ± 1.46 μm, *n* = 40. (**C**) NSCs of control (FRT82B), *hyx*$^{HT622}$, and *hyx*$^{W12-46}$ MARCM clones were labelled for α-tub and DNA and CD8-GFP. (**D**) Quantification of spindle length (with SD) observed in C. Spindle length: control, 9.70± 1.15 μm, *n* = 21; *hyx*$^{HT622}$, 6.13 ± 1.22 μm, *n* = 18; *hyx*$^{W12-46}$, 7.43 ± 1.25 μm, *n* = 18. (**E**) Ratio of spindle length to NSC diameter (with SD) for genotypes in A. Control: 0.82 ± 0.08, *n* = 32; *hyx* RNAi, 0.74 ± 0.12, *n* = 15. (**F**) Ratio of spindle length to NSC diameter (with SD) for genotypes in B. Control: 0.79 ± 0.06, *n* = 22; *hyx*$^{HT622}$, 0.70 ± 0.10, *n* = 13; *hyx*$^{W12-46}$, 0.70 ± 0.10, *n* = 21. (**G**) Interphase NSCs from control (FRT82B; *n* = 11), *hyx*$^{HT622}$ (*n* = 24), and *hyx*$^{w12-46}$ (*n* = 29) MARCM clones were labelled for α-tub, Asl, GFP, and PH3. NSCs failed to form a microtubule aster in 79.2% of *hyx*$^{HT622}$ and 34.5% of *hyx*$^{w12-46}$ clones. The rest of them formed weak microtubule asters. (**H**) The immunofluorescence intensity of astral microtubule (with SEM) labelled by α-tub for genotypes in G. Control: 100.7 ± 12.77, *n* = 7; *hyx*$^{HT622}$, 14.7 ± 19.12, *n* = 23; *hyx*$^{W12-46}$, 27.6 ± 32.56, *n* = 23. Cell outlines are indicated by white dotted lines. Arrows indicate the centrosomes. Statistical significances were determined by unpaired two-tailed Student *t* test in B. One-way ANOVA with multiple comparison were performed in D, F, and H. ****$p < 0.0001$ for B, D, F; **$p = 0.0065$ for E; In F, **$p = 0.0045$ for *hyx*$^{HT622}$ and **$p = 0.0017$ for *hyx*$^{W12-46}$. Scale bars: 5 μm. The underlying data for this figure can be found in the S1 Data. Asl, Asterless; α-tub, α-tubulin; Hyx, Hyrax; MARCM, mosaic analysis with a repressible cell marker; NSC, neural stem cell; RNAi, RNA interference.

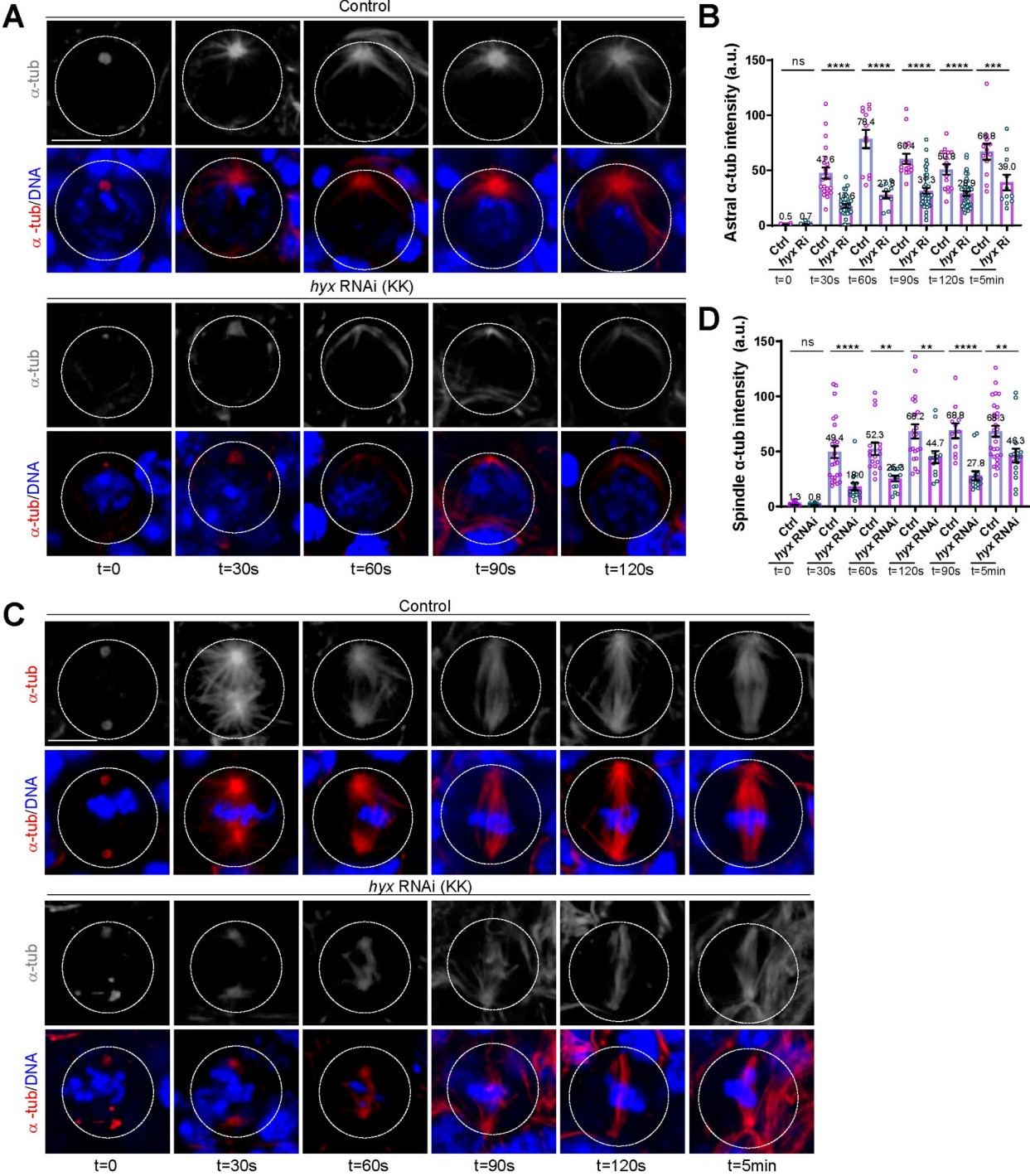

**Fig 4. Hyx promotes microtubule growth in NSCs.** (**A**) Control (*UAS-β-Gal* RNAi) and *hyx* RNAi (KK/V103555) interphase NSCs were labelled for α-tub and DNA at various time points after recovery from treatment on ice in microtubule regrowth assay. At 0 s (t = 0; ice treatment), no astral microtubules around the centrosome/MTOC were observed in all control (*n* = 74) and *hyx* RNAi NSCs (*n* = 63). At all subsequent time points following the recovery, all control NSCs assembled robust astral microtubules (t = 30 s, *n* = 31; t = 60 s, *n* = 11; t = 90 s, *n* = 27; t = 120 s, *n* = 26). Astral microtubules were lost in *hyx* RNAi: t = 30 s, 71.9%, *n* = 43; t = 60 s, 85.7%, *n* = 7; t = 90 s, 91.2%, *n* = 46; t = 2 min, 86.3%, *n* = 63. (**B**) The immunofluorescence intensity of astral microtubules (with SEM) marked by α-tub for various time points in A. Control: t = 0, 0.5 ± 0.13, *n* = 19; t = 30 s, 47.6 ± 5.01, *n* = 22; t = 60 s, 78.4 ± 0.13, *n* = 12; t = 90 s, 60.4 ± 4.58, *n* = 16; t = 2 min, 50.8 ± 4.65, *n* = 15; t = 5 min, 66.8 ± 7.27, n = 12. *hyx* RNAi: t = 0, 0.7 ± 0.22, *n* = 27; t = 30 s, 17.6 ± 1.73, *n* = 35; t = 60 s, 27.8 ± 3.27, *n* = 10; t = 90 s, 31.3 ± 2.60, *n* = 39; t = 2 min, 28.9 ± 2.04, *n* = 42; t = 5 min, 39.0 ± 7.07, *n* = 11. RNAi was controlled by *insc*-Gal4. (**C**) Control (*UAS-β-Gal* RNAi) and *hyx* RNAi (KK/V103555) metaphase NSCs were labelled for α-tub and DNA at various time points after recovery from treatment on ice in microtubule regrowth assay. Control: t = 0, *n* = 52;

t = 30 s, $n$ = 19; t = 60 s, $n$ = 9; t = 90 s, $n$ = 41; *hyx* RNAi, t = 0, $n$ = 28. Scarce spindle microtubule mass assembled in *hyx* RNAi: t = 30 s, 91.7%, $n$ = 18; t = 60 s, 66.7%, $n$ = 3; t = 90 s, 69.7%, $n$ = 19; t = 2 min, 85.4%, $n$ = 11. RNAi was controlled by *insc*-Gal4. (**D**) The immunofluorescence intensity of astral microtubules (with SEM) marked by α-tub for various time points in C. Control: t = 0, 1.4 ± 0.32, $n$ = 30; t = 30 s, 49.4 ± 5.53, $n$ = 26; t = 60 s, 52.3 ± 5.62, $n$ = 17; t = 90 s, 68.2 ± 6.43, $n$ = 21; t = 2 min, 68.8 ± 6.80, $n$ = 11; t = 5 min, 68.3 ± 4.97, $n$ = 28. *hyx* RNAi: t = 0, 0.8 ± 0.23, $n$ = 30; t = 30 s, 18.0 ± 3.43, $n$ = 15; t = 60 s, 25.3 ± 2.59, $n$ = 14; t = 90 s, 44.7 ± 5.52, $n$ = 13; t = 2 min, 27.8 ± 4.08, $n$ = 15; t = 5 min, 46.3 ± 6.18, $n$ = 16. Cell outlines are indicated by the white-dotted lines. One-way ANOVA with multiple comparison were performed in B and D. In B, ns > 0.9999, ****$p$ < 0.0001 and ***$p$ = 0.0003; in D, ns > 0.9999, ****$p$ < 0.0001, **$p$ = 0.0068 for t = 60 s, **$p$ = 0.0068 for t = 90 s, and **$p$ = 0.0034 for t = 5 min. Scale bars: 5 μm. The underlying data for this figure can be found in the S1 Data. α-tub, α-tubulin; Hyx, Hyrax; MTOC, microtubule-organizing center; NSC, neural stem cell; RNAi, RNA interference.

## Hyx is required for centrosome assembly in dividing NSCs

Since we had demonstrated that Hyx promotes microtubule growth and interphase aster formation in NSCs, we wondered whether Hyx regulates the assembly of the centrosomes, the major MTOC in dividing cells. Each centrosome is composed of a pair of centrioles surrounded by pericentriolar material (PCM) proteins. Centriolar proteins Spindle assembly abnormal 4 (Sas-4), Anastral spindle 2 (Ana2), and Asl are essential for centriole biogenesis and assembly [24,68–71]. In control interphase and metaphase NSCs, Sas-4 was always observed at the centrosomes overlapping with Asl (S5B and S5C Fig). In response to *hyx* knockdown, the localization of Sas-4 and Asl seemed unaffected at the centrosomes in both interphase and metaphase NSCs (S5B and S5C Fig).

Wild-type NSCs typically contained 2 Asl-positive centrioles at both interphase and metaphase (S5D Fig; $n$ = 21 and $n$ = 20, respectively), as the centrioles are duplicated early in the cell cycle. Interestingly, in *hyx*$^{HT622}$ MARCM clones, multiple centrioles labelled by Asl were seen at metaphase NSCs (S5D Fig; 28.1%, $n$ = 32) but not at interphase NSCs (S5D Fig; $n$ = 20). This observation suggests that centriole fragmentation might occur in *hyx*-depleted mitotic cells. Meanwhile, we found cytokinesis delay in 39.1% of *hyx*$^{HT622}$ clones (S5E Fig). However, no giant, polyploidy cells were observed for *hyx*-depleted brains, implying that cytokinesis delay is unlikely the primary cause of multiple centriole phenotype in these cells. Consistent with these observations, multiple centrioles were detected in 52.0% of S2 cells upon dsRNA treatment against *hyx* (S5F and S5G Fig), which was significantly higher than what was observed in control cells (S5F and S5G Fig; 25.1%).

Major PCM proteins γ-tubulin (γ-tub) and centrosomin (Cnn, a CDK5RAP2 homolog) are essential for microtubule nucleation and anchoring. γ-tub is a component of the γ-tubulin ring complex (γ-TURC), which is the major microtubule nucleator in dividing cells, including NSCs [72,73]. In control interphase and metaphase NSCs, robust γ-tub was detected at the centrosomes and was observed to colocalize with Asl (S6A Fig). By contrast, γ-tub was absent or significantly reduced at the centrosomes in 95.7% of *hyx*$^{HT622}$ and 54.8% of *hyx*$^{w12-46}$ interphase NSCs (S6A Fig); the fluorescence intensity of γ-tub was dramatically decreased in these NSCs (S6A and S6C Fig). In addition, γ-tub was strongly diminished at the centrosomes in 93.1% of *hyx*$^{HT622}$ and 70.8% of *hyx*$^{w12-46}$ metaphase NSCs (S6B Fig). The fluorescence intensity of γ-tub protein dropped to 0.17-fold in *hyx*$^{HT622}$ and 0.53-fold in *hyx*$^{w12-46}$ NSCs, respectively (S6B and S6C Fig). Likewise, γ-tub was strongly reduced at the centrosomes in 90.5% of interphase and 81.3% of metaphase NSCs upon *hyx* RNAi knockdown (S6D and S6E Fig); moreover, the fluorescence intensity of γ-tub was dramatically decreased at the centrosomes in these NSCs (S6D–S6F Fig).

Next, we examined the localization of Cnn, another essential PCM component. During interphase, 98.0% of control NSCs had intense Cnn signal at the centrosomes marked by Asl (S6G and S6I Fig). By contrast, Cnn was barely detectable at the centrosomes in 84.8% of *hyx*$^{HT622}$ and 58.3% of *hyx*$^{w12-46}$ ($n$ = 12) interphase NSCs (S6G and S6I Fig). Consistent with

this observation, Cnn levels were significantly diminished at the centrosomes in 72.4% of $hyx^{HT622}$ and 66.7% of $hyx^{w12-46}$ ($n = 18$) metaphase NSCs (S6H and S6I Fig). Similarly, Cnn levels were dramatically reduced from the centrosomes in 92.6% of interphase NSCs upon $hyx$ knockdown (S6J and S6L Fig). In metaphase NSCs with $hyx$ knockdown, Cnn intensity at the centrosomes significantly dropped in 46.7% of NSCs (S6K and S6L Fig). Our observations indicate that Hyx ensures the recruitment of PCM proteins γ-tub and Cnn to the centrosomes in both interphase and mitotic NSCs.

## Centrosomal localization of Msps, D-TACC, and Polo is dependent on Hyx function in NSCs

Msps is an XMAP215/ch-TOG family protein and a key microtubule polymerase that controls microtubule growth and asymmetric division of NSCs in *Drosophila* larval central brains [44,74]. In control interphase NSCs, Msps colocalized with Asl at the centrosomes (S7A Fig). However, during interphase, Msps was delocalized from the centrosomes in 86.4% of $hyx^{HT622}$ and 47.1% of $hyx^{w12-46}$ NSCs (S7A and S7C Fig). Likewise, during metaphase, Msps was nearly absent at the centrosomes in 87.5% of $hyx^{HT622}$ and 53.3% of $hyx^{w12-46}$ NSCs (S7B and S7C Fig). Moreover, Msps was undetectable at the centrosomes in the majority of interphase NSCs upon $hyx$ knockdown (S7D and S7F Fig). In metaphase NSCs, Msps was detected at the centrosomes only in 16.4% of $hyx$ RNAi NSCs (S7E and S7F Fig).

As the efficient centrosomal localization of Msps depends on D-TACC, a microtubule-binding centrosomal protein [74], we wondered whether D-TACC localization in NSCs requires Hyx function. In all control interphase NSCs, D-TACC was concentrated at the centrosomes marked by Asl (S7G Fig). Remarkably, during interphase, D-TACC was absent from the centrosomes in 92.0% of $hyx^{HT622}$ (S7G Fig) and 81.8% of $hyx^{W12-46}$ NSCs ($n = 22$). Similarly, in metaphase NSCs, D-TACC was undetectable at the centrosomes in 95.2% of $hyx^{HT622}$ NSCs (S7H Fig) and 80.0% of $hyx^{w12-46}$ NSCs ($n = 15$). Likewise, D-TACC was apparently undetectable at the centrosomes in the majority of interphase and metaphase NSCs upon $hyx$ knockdown (S7J and S7K Fig). The fluorescence intensity of D-TACC was significantly decreased in $hyx$-depleted NSCs (S7G–S7L Fig). Taken together, our results suggest that Hyx is essential for the centrosomal localization of Msps and D-TACC in cycling NSCs.

As Polo, another key centrosomal protein, is critical for the assembly of interphase microtubule asters and asymmetric cell division [27,75], we tested whether Hyx regulates the localization of Polo at the centrosomes. In control interphase NSCs, Polo was strongly detected at the centrosome marked by Asl (Fig 5A). In contrast, Polo was almost completely absent in 86.8% of $hyx^{HT622}$ and 58.8% of $hyx^{w12-46}$ interphase NSCs (Fig 5A). Furthermore, in control metaphase NSCs, Polo mainly appeared on the centrosomes and kinetochores and weakly on the mitotic spindle (Fig 5B). However, 80.0% of $hyx^{HT622}$ and 75% of $hyx^{w12-46}$ metaphase NSCs lost Polo loci, and the remaining NSCs only had a weak Polo signal (Fig 5B). Similarly, upon $hyx$ knockdown, Polo was almost completely lost from the centrosomes in 84.6% of interphase NSCs and 78.3% of metaphase NSCs (S8A and S8B Fig). The fluorescence intensity of Polo was significantly reduced at the centrosomes in $hyx$-depleted interphase and metaphase NSCs (Figs 5C and S8C).

Centrosomal protein AurA inhibits NSC overgrowth and regulates centrosome functions by directing the centrosomal localization of D-TACC and Msps [35,76]. We sought to examine whether the centrosomal localization of AurA is dependent on Hyx. AurA is clearly observed at the centrosomes marked by Asl in control interphase and metaphase NSCs (Fig 5D and 5E). Remarkably, AurA was nearly undetectable at the centrosomes in 87.2% of $hyx^{HT622}$ and 75.0% $hyx^{w12-46}$ interphase NSCs (Fig 5D). The fluorescence intensity of AurA decreased to

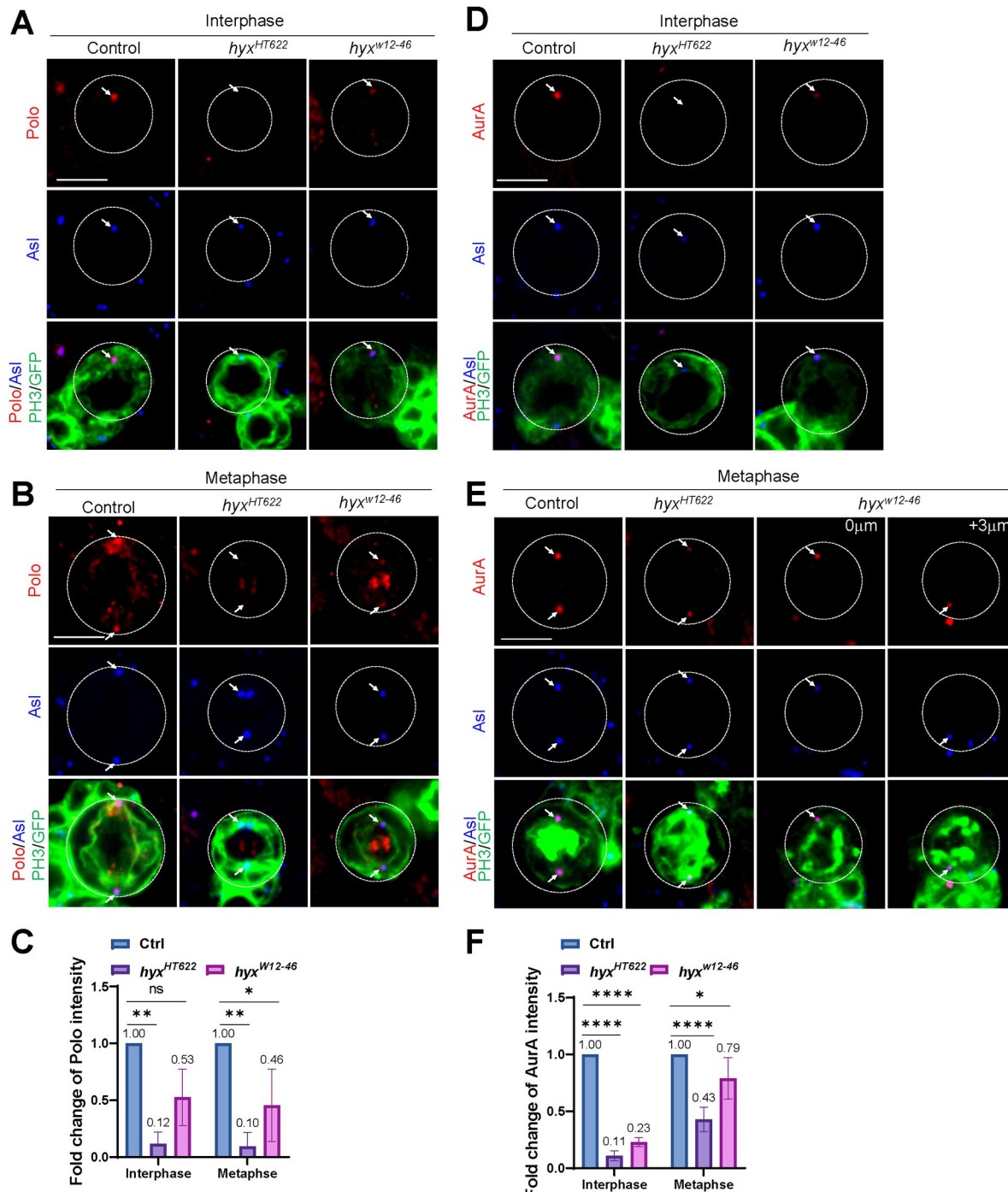

**Fig 5. Hyx regulates the localization of centrosomal proteins Polo and AurA in NSCs.** (**A**) Interphase NSCs from control (FRT82B) and $hyx^{HT622}$ MARCM clones were labelled for Polo, Asl, GFP, and PH3. Polo delocalization/reduction: control, 5.8%, $n = 52$; $hyx^{HT622}$, 86.8%, $n = 38$; $hyx^{w12-46}$, 58.8%, $n = 34$. (**B**) Metaphase NSCs of control (FRT82B), $hyx^{HT622}$ and $hyx^{w12-46}$ MARCM clones were labelled for Polo, Asl, GFP, and PH3. Polo delocalization/reduction: control, 0%, $n = 20$; $hyx^{HT622}$, 80.0%, $n = 10$; $hyx^{w12-46}$, 75.0%, $n = 12$. (**C**) Quantification graph of the fold change of Polo intensity (with SD) in NSCs from A and B. Interphase: control, 1-fold, $n = 41$; $hyx^{HT622}$, 0.12 ± 0.11-fold, $n = 38$; $hyx^{w12-46}$, 0.53 ± 0.25-fold, $n = 34$. Metaphase: control, 1-fold, $n = 20$; $hyx^{HT622}$, 0.10 ± 0.12-fold, $n = 10$; $hyx^{w12-46}$, 0.46 ± 0.32-fold, $n = 12$. (**D**) Interphase NSCs from control (FRT82B; $n = 36$) and $hyx^{HT622}$ ($n = 39$) MARCM clones were labelled for AurA, Asl, GFP, and PH3. (**E**) Metaphase NSCs of control (FRT82B; $n = 22$) and $hyx^{HT622}$ ($n = 18$) MARCM clones were labelled for AurA, Asl, GFP, and PH3. (**F**) Quantification graph of the fold change of AurA intensity (with SD) in NSCs from D and E. Interphase: control, 1-fold, $n = 36$; $hyx^{HT622}$, 0.11 ± 0.04-fold, $n = 39$; $hyx^{w12-46}$, 0.23 ± 0.04-fold, $n = 20$. Metaphase: control, 1-fold, $n = 22$; $hyx^{HT622}$, 0.43 ± 0.11-fold, $n = 18$; $hyx^{w12-46}$, 0.79 ± 0.18-fold, $n = 8$. Cell outlines are indicated by white-dotted lines. Arrows point at the centrosome in A-E. Statistical

significances were determined two-way ANOVA with multiple comparison were performed in C and F. In C, **$p$ = 0.0044, ns = 0.0634; in F, **$p$ = 0.0040, *$p$ = 0.0382. Scale bars: 5 μm. The underlying data for this figure can be found in the S1 Data. Asl, Asterless; AurA, Aurora-A; Hyx, Hyrax; MARCM, mosaic analysis with a repressible cell marker; NSC, neural stem cell.

0.11-fold in $hyx^{HT622}$ and 0.23-fold in $hyx^{w12-46}$ interphase NSCs (Fig 5D and 5F). Likewise, AurA levels were significantly reduced in 88.9% of $hyx^{HT622}$ and 50% of $hyx^{w12-46}$ metaphase NSCs (Fig 5E and 5F). Furthermore, AurA was diminished in 100% of interphase NSCs and 56.7% of metaphase NSCs upon $hyx$ knockdown (S8D–S8F Fig). Taken together, our data show that Hyx plays an essential role in centrosome assembly and functions by recruiting major centrosomal proteins to the centrosomes in NSCs.

## The disruption of NSC polarity and centrosome assembly is a direct consequence of *hyx* depletion, but not aging

To rule out the possibility that the disruption of NSC polarity and centrosome assembly was due to consequence of aging in late larval stages, we examined NSC polarity proteins and centrosomal proteins at 24 h ALH, a time point when NSCs exit quiescence and reenter the cell cycle [77]. At 24 h ALH, Hyx was dramatically lost in NSCs upon knocking down $hyx$ by RNAi under the control of *insc*-Gal4 driver in $hyx^{HT622/+}$ background, suggesting an efficient knockdown of Hyx (S9A Fig). In $hyx$ RNAi $hyx^{HT622/+}$ at 24 h ALH, aPKC was delocalized in 96.8% of NSCs and Mira in 91.9% of NSCs, compared with a control that both aPKC and Mira formed proper crescent in all metaphase NSCs (S9B Fig). Likewise, centrosome protein γ-tub was severely reduced at the centrosomes in 82.4% of interphase NSCs and 85.0% of metaphase NSCs from $hyx$ RNAi $hyx^{HT622/+}$ (S9C–S9E Fig). Similarly, Polo was largely delocalized from the centrosomes in NSCs from $hyx$ RNAi $hyx^{HT622/+}$, which led to a significant decrease of Polo protein levels at the centrosomes in both interphase and metaphase NSCs (S9F–S9H Fig).

In addition, in late larval stages, $hyx$ RNAi $hyx^{HT622/+}$ showed a stronger NSC overproliferation phenotype than that observed in $hyx$ knockdown alone (Fig 1B and 1C); 84.3% of type I lineages and 93.3% of type II lineages with multiple NSCs were observed in $hyx$ RNAi $hyx^{HT622/+}$ compared with the control with a single NSC per lineage (S10A Fig). Moreover, Hyx protein was diminished in 89.1% of $hyx$ RNAi $hyx^{HT622/+}$ NSCs, while it was strongly detected in the control (S10B Fig). Consistent with these observations, strong reduction of γ-tub and Polo protein levels at the centrosomes was observed in both interphase and metaphase NSCs from $hyx$ RNAi $hyx^{HT622/+}$ (S10C–S10H Fig).

Taken together, the disruption of NSC polarity and centrosome assembly is a direct consequence of $hyx$ loss of function instead of aging.

## Hyx is required for centrosome assembly in S2 cells in vitro

To investigate whether Hyx plays a role in centrosome assembly in nonneuronal cells, we knocked down $hyx$ in S2 cells by dsRNA treatment. We found that a centriolar protein, Ana2, remained localized at the centrosomes in metaphase cells (S8G Fig). This suggests that Hyx is not essential for the localization of centriolar proteins in both S2 cells and NSCs. Next, we examined the localization of other centrosomal proteins in S2 cells. Remarkably, D-TACC intensity was significantly decreased at the centrosomes, upon $hyx$ knockdown, in metaphase S2 cells (Fig 6A and 6B). Consistent with these observations, the intensity of α-tub was also decreased by 0.65-fold on mitotic spindles (Fig 6D and 6E). These in vitro data support our observations in the larval brain and indicate that Hyx regulates microtubule growth and the localization of centrosomal proteins. Polo is undetectable in interphase S2 cells, unlike its

robust localization in NSCs during the interphase. Consistent with our in vivo observations, we found that the overall intensity of Polo was significantly reduced to 0.67-fold in the dividing metaphase cells upon *hyx* knockdown (Fig 6A and 6C). Also, γ-tub intensity at the centrosomes marked by Ana2 was similar to that observed in the control (S8G and S8H Fig). The different observations in S2 cells and larval brains are likely due to incomplete depletion of *hyx* in S2 cells and/or different underlying mechanisms in vitro.

To further probe how Hyx regulates centrosome assembly, we examined the ultrastructure of Cnn and γ-tub using super-resolution imaging. Cnn and γ-tub formed "doughnut-like" rings surrounding the centriolar protein Asl, at the centrosomes, in 94.9% and 92.9% of control metaphase cells, respectively (Fig 6F and 6G). Remarkably, Cnn and γ-tub failed to form the ring patterns or formed a ring with reduced inner size at the centrosomes in 51.3% and 53.4% of *hyx* knockdown mitotic cells, respectively (Fig 6F and 6G). These observations suggest that Hyx is required for the proper recruitment of Cnn and γ-tub at the centrosomes in S2 cells.

## Hyx directly regulates the expression of *polo* and *aurA* in vitro

Next, we investigated whether Hyx directly regulates the expression of *polo* and *aurA*, the 2 key centrosomal proteins. We performed chromatin immunoprecipitation (ChIP) coupled with quantitative PCR (ChIP-qPCR) in S2 cells. After normalizing against "Pre-serum" (1-fold), only a 1.37-fold increase was seen for the negative control. In contrast, 2.94-fold enrichment was observed for *orb2* promoter, a positive control. Moreover, we found Hyx binds to the promoter region of *polo* (new Fig 6I; 2.63-fold and 2.95-fold using 2 pairs of primers). Hyx also binds to the promoter region of *aurA* (Fig 6I; 2.85-fold), but not numb (Fig 6I; 1.64-fold). Therefore, Hyx binds to the promoter region of both *polo* and *aurA*.

We performed the luciferase assay to verify the direct binding of Hyx to the *polo* promoter. The endogenous Hyx in S2 cells was able to induce the luciferase reporter activity under the control of *polo*-promoter (poloPro) normalized against Renilla luciferase activity, but not with the vector control (Fig 6J). We attempted to overexpress Hyx in S2 cells to test if it further enhances the luciferase activity under the control of the *polo* promoter. However, overexpression of Venus-tagged full-length *hyx* (*hyx*-FL) resulted in severe cell death (54.3%) detected by active Caspase-3 (S10I and S10J Fig; 11.2% cell death in the control), which precluded us from testing the effect of Hyx overexpression on the transcription of *polo* in the luciferase assay. Next, we sought to knock down *hyx* with dsRNA treatment in S2 cells and analyze the relative luciferase activity under the control of the *polo* promoter. The relative luciferase activity from the ds-*hyx* treatment group was significantly reduced to 0.5-fold compared with 1-fold from the control group (ds-*egfp*) (Fig 6K) The relative luciferase activity driven by *actin5c* promoter induced by ds-*hyx* treatment and ds-*egfp* groups was indistinguishable (Fig 6L; 1.0-fold versus 1.2-fold). We conclude that *hyx* can directly bind to the *polo* promoter region and promotes its transcription.

## The expression of centrosome-related genes depends on Hyx function in larval brains

As Parafibromin/Hyx regulates transcriptional events [78], we wondered whether Hyx was required for the expression of genes that are involved in centrosome assembly. To this end, we sought to perform reverse transcription quantitative real-time PCR (RT-qPCR) to detect differential transcription levels of those genes in larval brains. As both *hyx* alleles (*hyx*^HT622 and *hyx*^w12-46) led to embryonic lethality, we knocked down *hyx* in the larval brain by RNAi using a ubiquitous driver *actin5C-Gal4*. However, *hyx* RNAi under *actin5C*-Gal4 caused larval

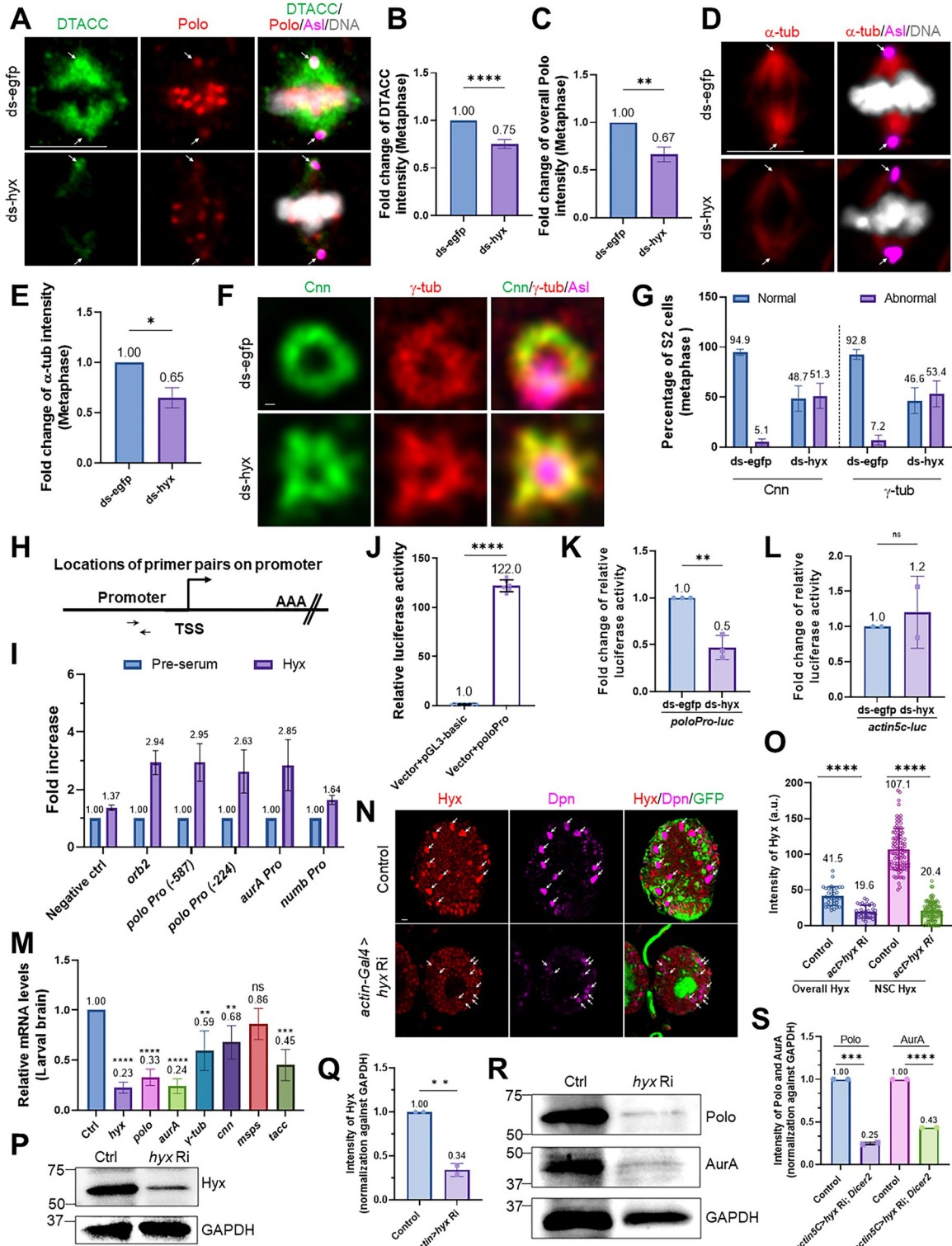

**Fig 6. Hyx is required for the recruitment of centrosome-related proteins and directly regulates their expression in vitro.** (**A**) Metaphase cells from ds-*egfp*-treated S2 cells and ds-*hyx*-treated S2 cells were labeled for DTACC, Polo, Asl, and DNA. (**B**) Quantification graph of the fold change of DTACC intensity (with SD) in S2 cells from A. ds-*egfp*, 1-fold, $n = 66$; ds-*hyx*, 0.75 ± 0.05-fold, $n = 69$. (**C**) Quantification graph of the fold change of overall Polo intensity (with SD) in S2 cells from A. ds-*egfp*, 1-fold, $n = 77$; ds-*hyx*, 0.67 ± 0.08-fold, $n = 81$. (**D**) Metaphase S2 cells treated with ds-*egfp* and ds-*hyx* were labeled for α-tub, Asl, and DNA. (**E**) Quantification graph of the fold change of α-tub intensity in S2 cells from D. ds-*egfp*, 1-fold, $n = 83$; ds-*hyx*, 0.65 ± 0.10-fold, $n = 88$. (**F**)

Spinning disc super-resolution imaging of Cnn, γ-tub, and Asl in metaphase S2 cells treated with ds-*egfp* (Cnn, *n* = 64; γ-tub, *n* = 55) and ds-*hyx* (Cnn, *n* = 58; γ-tub, *n* = 58). (**G**) Quantification graph of the percentage (with SD) of metaphase S2 cells forming "doughnut-like" shape of Cnn and γ-tub in F. ds-*egfp*: Pattern of Cnn, Normal, 94.9% ± 5.60%; abnormal, 5.1 ± 5.60%. ds-*hyx*: Pattern of Cnn, Normal, 48.7 ± 21.86%; abnormal, 51.3 ± 21.86%. ds-*egfp*: Pattern of γ-tub, Normal, 92.8 ± 8.57%; abnormal, 7.2 ± 8.57%. ds-*hyx*: Pattern of γ-tub, Normal, 46.6 ± 22.41%; abnormal, 53.4 ± 22.41%. (**H**) Location of primer pairs used for ChIP-qPCR on the promoter of genes in I. The schematic represents the *polo* gene, with the arrow indicating the upstream fragment distance from the TSS and the center nucleotide position of the primer pair is given and AAA showing the approximate location of the cleavage and polyadenylation site. (**I**) Quantification graph of ChIP-qPCR for detecting occupancy by Hyx on various genes in S2 cells, with an intergenic region at 5 kb downstream of the *numb* gene as a negative control and orb2 as a positive control. After normalizing against "Pre-serum," fold enrichment from "Pre-serum" was taken as 1-fold for all primer sets. Fold enrichment (with SD) in "Hyx": negative control, 1.37 ± 0.18-fold; positive control orb2, 2.94 ± 1.24-fold; *polo* (−587), 2.95 ± 1.47-fold; *polo* (−224), 2.63 ± 1.68-fold; *aurA*, 2.85 ± 1.77-fold; *numb*, 1.64 ± 0.31-fold. Minimum of 3 biological replicates were performed. (**J**) Luciferase assay in S2 cells shows an increase of the pGL3-luciferase reporter coupled with *polo* promoter (*poloPro*, 643 bp of sequences upstream of TSS) by endogenous *hyx* expression. Vector (pAFW)+pGL3-basic: 1.0 ± 0.28; Vector (pAFW)+ *poloPro*: 122 ± 6.05. The relative luciferase activity was normalized to Renilla luciferase activity. (**K**) Luciferase assay in S2 cells shows down-regulation of the pGL3-luciferase reporter coupled with *poloPro* ds-*hyx* treatment. The relative luciferase activity was normalized to Renilla luciferase activity. ds-*egfp*: 1-fold; ds-*hyx*: 0.5 ± 0.13-fold. (**L**) Luciferase assay in S2 cells shows no consistent alterations on the pGL3-luciferase reporter coupled with *actin5c* promoter between *ds-egfp* and ds-*hyx* treatment. The relative luciferase activity was normalized to Renilla luciferase activity. ds-*egfp*: 1-fold; ds-*hyx*: 0.2 ± 0.51-fold. (**M**) Quantification graph of RT-qPCR analysis in 48 h ALH brains from control (*UAS-β-gal* RNAi; *UAS-β-gal* RNAi) and *hyx* RNAi with *UAS-Dicer2* driven by *actin5C*-Gal4. Minimum 3 repeats were conducted. After normalization against control (with SD): control, 1-fold; *hyx*, 0.23 ± 0.05-fold; *polo*, 0.33 ± 0.08-fold; *aurA*, 0.24 ± 0.07-fold; γ-*tub*, 0.59 ± 0.20-fold; *cnn*, 0.68 ± 0.17-fold; *msps*, 0.86 ± 0.15-fold; *tacc*, 0.45 ± 0.15-fold. (**N**) 48 h ALH larval brains from control *(UAS-β-Gal* RNAi; *UAS-β-Gal* RNAi) and *hyx* knockdown with *UAS-Dicer2* (*hyx* RNAi; *UAS-Dicer2* RNAi) under the control of *actin5C*-Gal4 were labelled with Hyx, Dpn, and Phalloidin. (**O**) Quantification graph for (N) showing the fluorescence intensity of Hyx throughout the whole brain and in the NSCs, respectively. Overall Hyx intensity in control: 41.5 ± 13.25, *n* = 34 ROI (region of interest excluded neuropil region) from 10 BL; *hyx* RNAi: 19.7 ± 8.99, *n* = 36 ROI from 8 BL. Hyx intensity in NSCs in control: 107.1 ± 29.18, *n* = 96 NSCs; *hyx* RNAi: 20.4 ± 12.92, *n* = 99 NSCs. (**P**) Western blotting analysis of larval brain protein extracts of control (*UAS-β-Gal* RNAi; *UAS-β-Gal* RNAi) and *hyx* knockdown with *UAS-Dicer2* (*hyx* RNAi; *UAS-Dicer2* RNAi) driven by *actin5C*-Gal4 at 48 h ALH. Blots were probed with anti-Hyx antibody and anti-GAPDH antibody. (**Q**) Fold change of Hyx protein after normalization against GAPDH in brain extracts from control (*UAS-β-gal* RNAi; *UAS-β-gal* RNAi) and *hyx* RNAi with *UAS-Dicer2* driven by *actin5C*-Gal4 at 48 h ALH. Control, 1-fold; *hyx* RNAi, 0.34 ± 0.07-fold. (**R**) Western blotting analysis of 48 h ALH larval brain extracts of control (*UAS-β-Gal* RNAi; *UAS-β-Gal* RNAi) and *hyx* knockdown with *UAS-Dicer2* (*hyx* RNAi; *UAS-Dicer2* RNAi) under the control of *actin5C*-Gal4. Blots were probed with anti-Polo antibody, anti-AurA antibody, and anti-GAPDH antibody. (**S**) Fold change of Polo and AurA protein intensity after normalization against GAPDH in brain extracts from control (*UAS-β-gal* RNAi; *UAS-β-gal* RNAi) and *hyx* RNAi with *UAS-Dicer2* driven by *actin5C*-Gal4 at 48 h ALH. Polo: control, 1-fold; *hyx* RNAi, 0.25 ± 0.01-fold. AurA: control, 1-fold; *hyx* RNAi, 0.43 ± 0.00-fold. Two individual repeats were conducted (P and R). Error bars indicate standard deviation in J-S. Arrows indicate the centrosomes in A, D and NSCs in N. Statistical significances were determined by unpaired two-tailed Student *t* test in B, C, E, K-M, O, Q, and S. ****$p < 0.0001$ for B, J, M, O, and S; ***$p = 0.0004$ for *tacc* in M;**$p = 0.0017$ for C, **$p = 0.0063$ for γ-*tub* in M, and **$p = 0.0084$ for *cnn* in M; *$p = 0.0391$ for E; D, F; **$p = 0.0065$ for E; **$p = 0.0020$ for K; ns = 0.6287 for L; ns = 0.1224 for *msps* in M; **$p = 0.0036$ for Q; ***$p = 0.0002$ for S. Scale bars: 5 μm. The underlying data for this figure can be found in the S1 Data. ALH, after larval hatching; Asl, Asterless; AurA, Aurora-A; ChIP-qPCR, chromatin immunoprecipitation coupled with quantitative PCR; Cnn, centrosomin; Hyx, Hyrax; msps, mini spindles; NSC, neural stem cell; RNAi, RNA interference; ROI, region of interest; RT-qPCR, reverse transcription quantitative real-time PCR; TSS, transcription start site; α-tub, α-tubulin; γ-tub, γ-tubulin.

lethality after 48 h ALH. Therefore, we performed both RT-qPCR and western blot experiments for *hyx* RNAi under the control of *actin5C*-Gal4 at 48 h ALH. *hyx* mRNA levels were dramatically reduced to 0.23-fold upon *hyx* knockdown (Fig 6M). The overall Hyx throughout *hyx*-depleted larval brains was significantly reduced (fluorescence intensity 19.7 ± 8.99 a.u, *n* = 8 BL), compared with that in control (41.5 ± 13.25, *n* = 10 BL) (Fig 6N and 6O). Particularly, Hyx levels were dramatically decreased in these *hyx*-depleted brains (20.4 ± 12.92, *n* = 99 NSCs), compared with that in control (107.1 ± 29.18, *n* = 96 NSCs) (Fig 6N and 6O).

Remarkably, mRNA levels of *polo* and *aurA* were dramatically decreased to 0.33-fold and 0.24-fold, respectively, following *hyx* knockdown under *actin5C*-Gal4 (Fig 6M). Meanwhile, various centrosomal genes were significantly down-regulated upon *hyx* depletion in the larval brain (Fig 6M; γ-*tub*, 0.59-fold; *cnn*, 0.68-fold; *tacc*, 0.45-fold), but not *msps* (0.86-fold). Among cell polarity genes, only *par6* was significantly decreased to 0.68-fold (S10K Fig), and *baz* (0.84-fold), *pins* (1.04-fold), *insc* (1.07-fold), and *numb* (1.81-fold) were unaffected. By contrast, aPKC mRNA levels were not significantly affected (S10K Fig, 5.21-fold). The high

variation of aPKC mRNA level might be due to unstable aPKC mRNA at the time point of analysis (48 h ALH) when larvae started dying.

Moreover, the western blot result showed that Hyx protein levels were dramatically reduced in *hyx* RNAi brain under *actin5C*-Gal4 to 0.34-fold in contrast to 1-fold in control (Fig 6P and 6Q), indicating an efficient *hyx* knockdown using *actin5C*-Gal4. Both Polo (0.25-fold) and AurA (0.43-fold) levels normalized against GAPDH levels were dramatically decreased following *hyx* depletion, compared with 1-fold in control (Fig 6R and 6S).

Taken together, our data suggest that Hyx appears to primarily regulate the expression of *polo* and *aurA* in NSCs.

## *polo* and *aurA* are 2 key downstream targets of Hyx in NSCs

Given that Polo and AurA regulate cell polarity and microtubule functions in NSCs, we sought to investigate whether Polo and AurA are physiologically relevant targets of Hyx in NSCs. We overexpressed *polo* and *aurA* in the *hyx* RNAi knockdown background and found that the ectopic NSC phenotype caused by *hyx* depletion was significantly suppressed. With the introduction of *Venus-polo* and *aurA* into *hyx* RNAi, the average total NSC number of each brain lobe was significantly reduced to 111.1 ± 10.1 and 114.0 ± 5.1, respectively (Fig 7A and 7B; *n* = 7 BL and *n* = 11 BL, respectively) compared with 133.8 ± 4.0 NSCs in *hyx* RNAi alone (Fig 7A and 7B; *n* = 5 BL), close to 98.5 ± 2.8 in control larval brains (Fig 7A and 7B; *n* = 11 BL), Venus-Polo- (99.2 ± 2.4, *n* = 10 BL) or AurA-overexpressing brains (98.9 ± 3.6, *n* = 12 BL). Moreover, the average type I NSC number per lineage in *hyx* RNAi with *Venus-polo* and *aurA* overexpression was decreased to 1.7 ± 0.13 (*n* = 66 NSC lineage) and 2.3 ± 0.32 (*n* = 27 NSC lineage) (Fig 7C; mean ± SEM), significantly lower than 3.9 ± 0.77 (*n* = 27 NSC lineage) in *hyx* RNAi with *β-gal* RNAi. Likewise, the average type II NSC number per lineage in *hyx* RNAi with *polo* and *aurA* overexpression was significantly dropped to 4.6 ± 0.40 (*n* = 36 NSC lineage) and 4.5 ± 0.50 (*n* = 21 NSC lineage), respectively, compared with 7.7 ± 1.4 (*n* = 19 NSC lineage) in *hyx* RNAi with *β-gal* RNAi. Only 1 NSC per lineage was observed in both type I and type II NSC lineages from *β-gal* RNAi control, *Venus-polo* overexpression control, and *aurA* overexpression control.

Importantly, cell polarity defects caused by *hyx* depletion were also significantly suppressed by overexpression of *polo* and *aurA* in NSCs. The majority (73.4%) of *hyx* RNAi metaphase NSCs had lost aPKC polarity and the remaining 26.6% of NSCs had a weak aPKC crescent (Fig 7D and 7E; *n* = 55). Remarkably, 4.1% of *hyx* RNAi NSCs with *Venus-polo* overexpression formed a strong aPKC crescent and 48.9% showed a weak aPKC crescent (Fig 7D and 7E; *n* = 59 NSC). Similarly, a strong Mira crescent was observed in 1.4% of *hyx* RNAi NSCs with *Venus-polo* overexpression and 57.0% formed a weak Mira crescent (Fig 7D and 7F; *n* = 59), compared with 29.8% of NSCs with a weak Mira crescent in *hyx* knockdown alone (Fig 7D and 7F; *n* = 55). Likewise, *aurA* overexpression can dramatically restore the asymmetric localization of both aPKC and Mira in *hyx* RNAi NSCs. Upon *aurA* overexpression, a strong aPKC crescent was seen in 6.5% of *hyx* RNAi NSCs and a weak aPKC crescent was formed in 54.7% of NSCs (Fig 7D and 7E; *n* = 51 NSC). Similarly, upon *aurA* overexpression, 6.5% of *hyx* RNAi NSCs formed a strong Mira crescent and 51.5% showed a weak Mira crescent (Fig 7D and 7F; *n* = 51 NSC). These suppressions were partial because the vast majority of metaphase NSCs from control (*n* = 58), Venus-Polo- (*n* = 25), or AurA-overexpression (*n* = 24) assembled strong aPKC and Mira crescent on the cortex (Fig 7D). Moreover, the shorter spindle phenotype in *hyx* RNAi was also significantly suppressed by *polo* and *aurA* overexpression (Fig 7G and 7H; control, 0.90 ± 0.05, *n* = 18 NSCs; 0.82 ± 0.06, *n* = 23 NSCs; 0.92 ± 0.06, *n* = 22 NSCs; 0.91 ± 0.06, *n* = 27 NSCs). These results support our conclusion that both Polo and AurA are

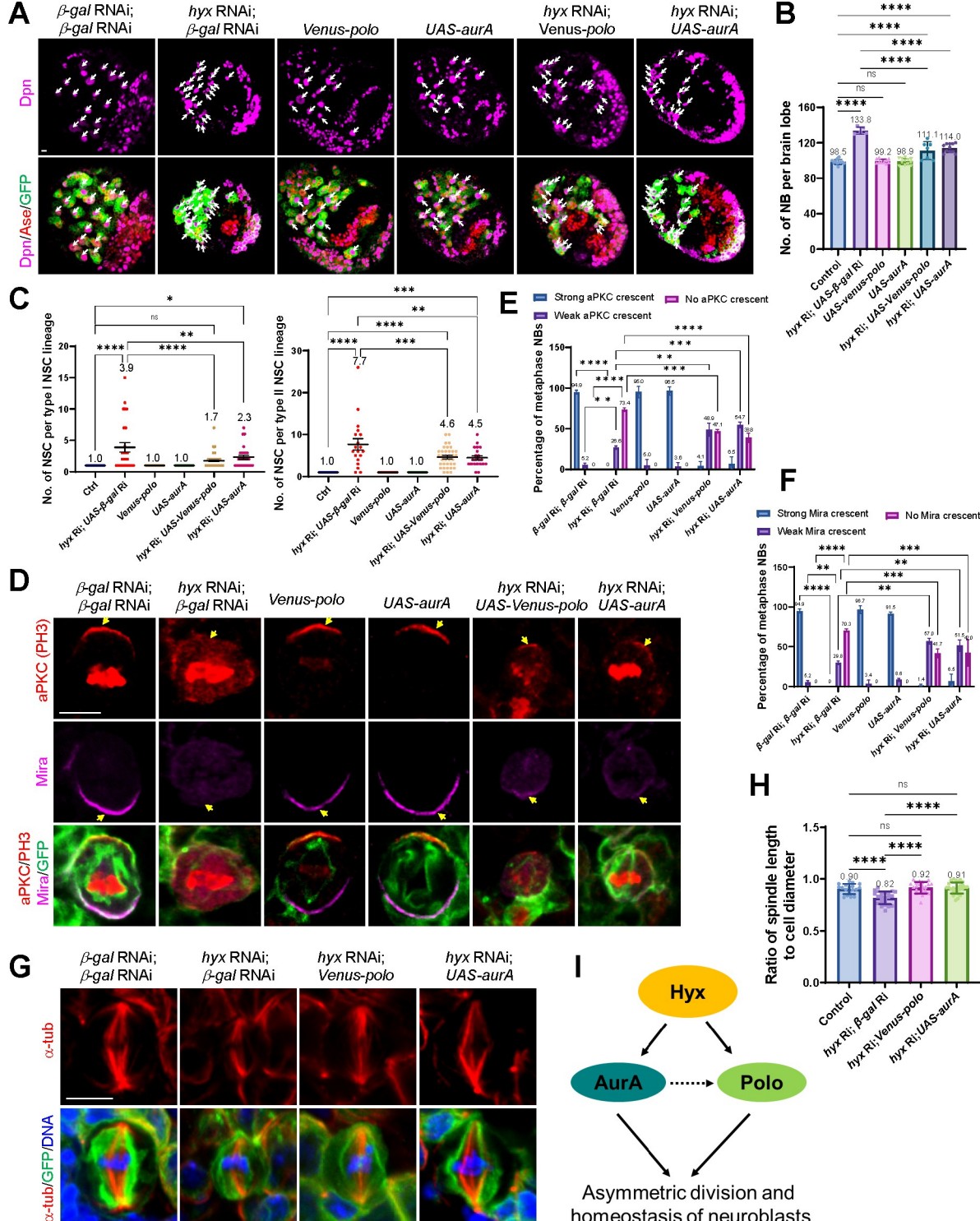

**Fig 7. Polo and AurA are 2 key downstream targets of Hyx and functional for preventing *hyx*-depletion mediated NSC overproliferation.** (**A**) Third instar larval brains from control *(UAS-β-Gal* RNAi; *UAS-β-Gal* RNAi), *hyx* knockdown with UAS control (*hyx* RNAi; *UAS-β-Gal* RNAi), *hyx* knockdown with *polo* expression (*hyx* RNAi; *UAS-venus-polo*) and *hyx* knockdown with *aurA* expression (*hyx* RNAi; *UAS-aurA*) were labelled with Dpn, Ase, and GFP. RNAi was driven by *insc*-Gal4. (**B**) Average NSC number per brain lobe (with SD) for genotypes in A. Control: 98.5 ± 2.4 NSC, *n* = 11 BL; *hyx* RNAi; *UAS-β-Gal* RNAi: 133.8 ± 4.01 NSC, *n* = 5 BL; *venus-polo*: 99.2 ± 2.4, *n* = 10 BL; *UAS-aurA*: 98.9 ± 3.6, *n* = 12 BL; *hyx* RNAi; *UAS-venus-polo*: 11.1 ± 10.1, *n* = 7 BL; *hyx* RNAi; *UAS-aurA*: 114.0 ± 5.1 NSC, *n* = 11 BL. (**C**) Average NSC number per lineage (with SEM) for genotypes in A. Type I: Control, 1.0 ± 0 NSC, *n* = 20 NSC lineage; *hyx*

RNAi with *UAS-β-Gal* RNAi, 3.9 ± 0.77 NSC, $n$ = 27 NSC lineage; *venus-polo*, 1.0 ± 0, $n$ = 34 NSC lineage; *UAS-aurA*, 1.0 ± 0, $n$ = 36 NSC lineage; *hyx* RNAi with *UAS-venus-polo*, 1.7 ± 0.13, $n$ = 66 NSC lineage; *hyx* RNAi with *UAS-aurA*, 2.3 ± 0.32 NSC, $n$ = 27 NSC lineage. Type II: Control, 1.0 ± 0 NSC, $n$ = 20 NSC lineage; *hyx* RNAi with *UAS-β-Gal* RNAi, 7.7 ± 1.4 NSC, $n$ = 19 NSC lineage; *venus-polo*, 1.0 ± 0, $n$ = 22 NSC lineage; *UAS-aurA*, 1.0 ± 0, $n$ = 24 NSC lineage; *hyx* RNAi with *UAS-venus-polo*, 4.6 ± 0.40, $n$ = 36 NSC lineage; *hyx* RNAi with *UAS-aurA*, 4.5 ± 0.50 NSC, $n$ = 21 NSC lineage. (**D**) Metaphase NSCs from third instar larval brains of control (*UAS-β-Gal* RNAi; *UAS-β-Gal* RNAi), *hyx* knockdown with UAS control (*hyx* RNAi; *UAS-β-Gal* RNAi), *hyx* knockdown with *polo* expression (*hyx* RNAi; *UAS-venus-polo*) and *hyx* knockdown with *aurA* expression (*hyx* RNAi; *UAS-aurA*) were stained with aPKC, Mira, PH3, and GFP. RNAi was controlled by *insc*-Gal4. (**E**) Percentage of NSC with indicated levels of aPKC crescent for genotypes in D. Control ($n$ = 58 NSC): Strong, 94.9%; Weak, 5.2%; No, 0; *hyx* knockdown with UAS control ($n$ = 55 NSC): Strong, 0; Weak, 26.6%; No, 73.4%; *venus-polo* ($n$ = 25 NSC): Strong, 95.0%; Weak, 5.0%; No, 0; *UAS-aurA* ($n$ = 24 NSC): Strong, 96.5%; Weak, 3.6%; No, 0; *hyx* knockdown with *polo* expression ($n$ = 59 NSC): Strong, 4.1%; Weak, 48.9%; No, 47.1%; *hyx* knockdown with *aurA* expression ($n$ = 51 NSC): Strong, 6.5%; Weak, 54.7%; No, 38.8%. (**F**) Percentage of NSC with indicated levels of Mira crescent for genotypes in D. Control ($n$ = 58 NSC): Strong, 94.9%; Weak, 5.2%; No, 0; *hyx* knockdown with UAS control ($n$ = 55 NSC): Strong, 0; Weak, 29.8%; No, 70.3%; *venus-polo* ($n$ = 25 NSC): Strong, 96.7%; Weak, 3.4%; No, 0; *UAS-aurA* ($n$ = 24 NSC): Strong, 91.5%; Weak, 8.6%; No, 0; *hyx* knockdown with *polo* expression ($n$ = 59 NSC): Strong, 1.4%; Weak, 57.0%; No, 41.7%; *hyx* knockdown with *aurA* expression ($n$ = 51 NSC): Strong, 6.5%; Weak, 51.5%; No, 42.0%. (**G**) Metaphase NSCs of control (*UAS-β-Gal* RNAi; *UAS-β-Gal* RNAi), *hyx* knockdown with UAS control (*hyx* RNAi; *UAS-β-Gal* RNAi), *hyx* knockdown with *polo* expression (*hyx* RNAi; *UAS-venus-polo*) and *hyx* knockdown with *aurA* expression (*hyx* RNAi; *UAS-aurA*) under the control of *insc*-Gal4 were labelled for α-tub, GFP, and DNA. (**H**) Ratio of spindle length to NSC diameter (with SD) for genotypes in G. Control: 0.90 ± 0.05, $n$ = 18; *hyx* knockdown with UAS control: 0.82 ± 0.06, $n$ = 23; *hyx* knockdown with *polo* expression: 0.92 ± 0.06, $n$ = 22; *hyx* knockdown with *aurA* expression: 0.91 ± 0.06, $n$ = 27. (**I**) A working model. One-way ANOVA with multiple comparison was performed in B, C and H. $^{****}p < 0.0001$ in B, C and H; ns > 0.9999 in B; ns = 0.9647 and 0.9874 in H. For type I group from C: $^{**}p = 0.0035$, $^{*}p = 0.0350$, and ns = 0.3465. For type II group from C: $^{***}p = 0.0005$, $^{***}p = 0.0003$, and $^{**}p = 0.0012$. Two-way ANOVA with multiple comparison was performed in E and F. $^{****}p < 0.0001$ in E and F; for "Weak aPKC crescent" in E: $^{**}p = 0.0028$; $^{***}p = 0.0002$. For "No aPKC crescent" in E: $^{***}p = 0.0003$. For "Weak Mira crescent" in F: $^{**}p = 0.0029$, $^{**}p = 0.0010$, and $^{**}p = 0.0089$. For "No Mira crescent" in F: $^{***}p = 0.0006$, and $^{***}p = 0.0007$. Scale bars: 5 μm. The underlying data for this figure can be found in the S1 Data. aPKC, atypical PKC; AurA, Aurora-A; Hyx, Hyrax; Mira, Miranda; NSC, neural stem cell; RNAi, RNA interference; α-tub, α-tubulin.

physiologically relevant targets of Hyx in NSCs. Therefore, down-regulated *polo* and *aurA* expression likely accounts for various defects, such as loss of asymmetry of apical and basal proteins and centrosome/microtubule abnormalities, observed in *hyx*-depleted NSCs. Polo can be directly activated by AurA in *Drosophila* mitotic NSCs [79]. Our data support a model in which Hyx promotes the expression of *polo* and *aurA* in NSCs and, in turn, regulates cell polarity and centrosome/microtubule assembly (Fig 7I).

**Discussion.** In this study, we established the essential role of Hyx, the *Drosophila* ortholog of Parafibromin, during the development of the CNS. We show that Hyx governs NSC asymmetric division and inhibits ectopic NSC formation in the central brains of *Drosophila* larvae. We also demonstrate that Hyx plays a novel function in the formation of microtubule asters and mitotic spindles in interphase NSCs. Particularly, Hyx is important for the localization of PCM proteins to the centrosomes in dividing NSCs and S2 cells. Therefore, this is the first study to demonstrate that Parafibromin/Hyx plays a critical role in the asymmetric division of NSCs, by maintaining NSC polarity and regulating microtubule/centrosomal assembly in these cells.

It is established that *Drosophila* Hyx is essential for embryogenesis and wing development [51]. In this study, we provide the first evidence that Hyx is crucial for *Drosophila* larval brain development. Furthermore, we showed that Hyx is essential for the polarized distribution of proteins in dividing NSCs, indicating a novel role for Hyx in regulating NSC apicobasal polarity. We also found that Hyx is required for the centrosomal localization of AurA and Polo kinases in NSCs, 2 brain tumor suppressor-like proteins that regulate asymmetric cell divisions [26–28]. Mechanistically, Hyx promotes the expression of both *aurA* and *polo* by directly binding to the promoter region of these 2 genes. Hyx does not seem to affect the expression of a few other polarity genes we have tested, such as *apkc*, *baz*, *pins*, and *numb*. Overexpression of either *polo* or *aurA* partially suppressed ectopic NSC formation and NSC polarity defects caused by *hyx* depletion, suggesting that *polo* or *aurA* are 2 physiological relevant targets of Hyx in NSCs. Therefore, our study identifies a previously unknown link between Hyx and

these cell cycle regulators, raising an interesting possibility that similar regulatory mechanisms may exist in other types of dividing cells, including cancer cells.

Human Parafibromin is a well-known tumor suppressor in parathyroid carcinomas and many other types of cancers [45–48]. Parafibromin primarily regulates transcriptional events and histone modifications [49,50]. It is also known to inhibit cell proliferation by the blockage of a G1 cyclin, Cyclin D1, and the c-myc proto-oncogene [51,78,80]. We show that Hyx controls *polo* expression likely through a direct transcriptional regulation. Among Paf1 complex components, only *rtf1* RNAi resulted in weak ectopic type II NSCs (but no ectopic type I NSCs). This finding suggests that the function of Hyx in asymmetric cell division of NSCs is largely independent of other components of the Paf1 complex. As our data support the role of Hyx in transcriptional regulation, it appears that Hyx alone without other Paf1 components might be sufficient to promote the target gene transcription. Nevertheless, we cannot rule out the possibility that other Paf1 components are involved in the asymmetric division to a much lesser extent or have redundant functions.

In addition to its role in promoting asymmetric cell divisions and the establishment of api-cobasal cell polarity, we provide compelling data that Hyx plays a novel role in regulating microtubule growth and centrosomal assembly in NSCs and S2 cells. Hyx was found to be important for the formation of interphase microtubule asters and the mitotic spindle. We also showed that it is required for the centrosomal localization of major PCM proteins in NSCs, including γ-tub, Cnn, AurA, and Polo. AurA is known to recruit γ-TuRC, Cnn, and D-TACC to the centrosomes [76]. Centrosomal proteins, such as Msps/D-TACC and Cnn, may not be well recruited due to defective centrosome maturation and not a direct effect of Hyx on their expression. Unlike in dividing neuroblasts, in S2 cells, γ-tubulin is important for the nucleation of both centrosomal microtubules and noncentrosomal microtubules, i.e., chromatin-mediated microtubule assembly [81]. Therefore, in S2 cells, even with a reduction of Polo and AurA on the centrosomes, γ-tubulin might be recruited to the spindle poles in a centrosome-independent manner. The new role for Hyx in regulating microtubule growth and asymmetric divisions in NSCs proposed in this study is consistent with our previous finding that NSC polarity is dependent on microtubules [44].

In addition to its predominant localization and functions in the nucleus, Parafibromin is also known to exist in the cytoplasm, where it regulates apoptosis by directly targeting p53 mRNA [82]. Human Parafibromin directly interacts with actin-binding proteins, actinin-2 and actinin-3, during the differentiation of myoblasts [83], suggesting that Parafibromin might regulate the actin cytoskeleton. Interestingly, *C. elegans* Ctr9 is required for the microtubule organization in epithelial cells during the morphogenesis of the embryo [84]. Therefore, the function of Hyx/Parafibromin in regulating centrosomal assembly is likely a general paradigm in cell division regulation, which might be disrupted in cancer cells. Parafibromin/HRPT-2 is expressed in both mouse and human brains [52]. Deletion of *Hrpt2* in mouse embryos results in early lethality and a developmental defect of the brain, suggesting that Parafibromin may play a role in CNS development [85]. Further investigations on the likely conserved functions of mammalian Parafibormin in NSC divisions and microtubule growth are warranted in future studies.

## Materials and methods

### Fly stocks and genetics

Fly stocks and genetic crosses were reared at 25˚C unless otherwise stated. Fly stocks were kept in vials or bottles containing standard fly food (0.8% *Drosophila* agar, 5.8% Cornmeal, 5.1% Dextrose, and 2.4% Brewer's yeast). The following fly strains were used in this study: *insc*-Gal4

(BDSC#8751; 1407-Gal4), *insc*-Gal4, *UAS-Dicer2* with and without *UAS-CD8-GFP*, *Jupiter-GFP* (G147), UAS-*hyx*, *UAS-HRPT2* [51]. The type II NSC driver: w; *UAS-Dicer 2*, *wor*-Gal4, *ase*-Gal80/CyO; *UAS-mCD8-GFP*/TM3, Ser [86]; *hyx* RNAi *hyx^{HT622}*, *UAS-venus-polo* [28].

The following stocks were obtained from Bloomington Drosophila Stock Center (BDSC): *UAS-Gal* RNAi (BDSC#50680; this stock is often used as a control UAS element to balance the total number of UAS elements), *ctr9^{12P023}* (BDSC#59389), *rtf1* RNAi (BDSC#36586), *rtf1* RNAi (BDSC#34850) [87], *rtf1* RNAi (BDSC#31718), *UAS-aurA* (BDSC#8376), *actin5C*-Gal4 (BDSC#25374).

The following stocks were obtained from Vienna Drosophila Resource Center (VDRC): *hyx* RNAi (28318), *hyx* RNAi (103555), *atms* RNAi (108826) [87], *atu* RNAi (17490) [88], *atu* RNAi (106074), *ctr9* RNAi (108874), *ctr9* RNAi [89], *rtf1* RNAi (27341) [88], *rtf1* RNAi (110392).

All experiments were carried out at 25°C, except for RNAi knockdown or overexpression experiments that were performed at 29°C.

## Immunohistochemistry

Third instar *Drosophila* larvae were dissected in PBS, and larval brains were fixed in 4% EM-grade formaldehyde in PBT (PBS + 0.3% Triton-100) for 22 min. The samples were processed for immunostaining as previously described [77]. For α-tubulin immunohistochemistry, larvae were dissected in Shield and Sang M3 medium (Sigma-Aldrich), supplemented with 10% FBS, followed by fixation in 10% formaldehyde in Testis buffer (183 mM KCl, 47 mM NaCl, 10 mM Tris-HCl, and 1 mM EDTA (pH 6.8)), supplemented with 0.01% Triton X-100. The fixed brains were washed once in PBS and twice in 0.1% Triton X-100 in PBS. Images were taken with an AxioCam HR camera (with 1.5× to 8× of digital zoom) of a LSM710 confocal microscope system (Axio Observer Z1; ZEISS), using a Plan-Apochromat 40×/1.3 NA oil differential interference contrast objective. The brightness and contrast of the images obtained were adjusted using Adobe Photoshop or Fiji (imageJ).

The primary antibodies used were the following: rabbit affinity-purified anti-Hyx/Cdc73 (1:1,000; J. T. Lis); guinea pig anti-Dpn (1:1,000), mouse anti-Mira (1:50; F. Matsuzaki), rabbit anti-Mira (1:500, W. Chia), anti-Insc (1:1,000, X. Yang), rabbit anti-aPKCζ C20 (1:100; Santa Cruz Biotechnologies, Dallas, TX), guinea pig anti-Numb (1:1,000; J Skeath), rabbit anti-GFP (1:3,000; F. Yu), mouse anti-GFP (1:5,000; F. Yu), rabbit anti-Asense (1:1,000; YN Jan), guinea pig anti-Asl (1:200, C. Gonzalez), rabbit anti-Sas-4 (1:100, J. Raff), mouse anti-α-tubulin (1:200, Sigma, Cat#: T6199), mouse anti-γ-tubulin (1:200, Sigma, Cat#: T5326), rabbit anti-Cnn (1:5,000, E. Schejter and T. Megraw), rabbit anti-Msps (1:500), rabbit anti-Msps (1:1,000, J. Raff), rabbit anti-PH3 (1:200, Sigma, Cat#: 06–570), rabbit anti-DTACC (1:200), rabbit anti-Ana2 [24], α-tubulin (1:200, Sigma, Cat#: T6199), rabbit anti-AurA (1:200, J. Raff), rat anti-CD8 (1:250, Caltag Laboratories), mouse anti-Polo (1:30, C. Sunkel), rabbit anti-Cleaved Caspase-3 (Asp175) (1:100, Cell Signaling, Cat#: 9664), rabbit anti-Phospho Drosophila Akt (Ser505) (1:100, Cell Signaling, Cat#: 4054), rabbit anti-Stg/Cdc25 (1:500, Eric F. Wieschaus), rabbit anti-Mts (1:50). The secondary antibodies used were conjugated with Alexa Fluor 488, 555, or 647 (Jackson Laboratory).

## Spinning disc super-resolution imaging

Super-resolution Spinning Disc Confocal-Structured Illumination Microscopy (SDC-SIM) was performed on a spinning disk system (Gataca Systems) based on an inverted microscope (Nikon Ti2-E; Nikon) equipped with a confocal spinning head (CSU-W; Yokogawa), a Plan-Apo objective (100×1.45-NA), and a back-illuminated sCMOS camera (Prime95B; Teledyne

Photometrics). A super-resolution module (Live-SR; GATACA Systems) based on structured illumination with optical reassignment technique and online processing leading to a 2-time resolution improvement [90] is included. The maximum resolution is 128 nm with a pixel size of 64 nm in super-resolution mode. Excitation light at 488 nm/150 mW (Vortran) (for GFP), 561 nm/100 mW (Coherent) (for mCherry/mRFP/tagRFP) and 639 nm/150 mW (Vortran) (for iRFP) was provided by a laser combiner (iLAS system; GATACA Systems), and all image acquisition and processing were controlled by the MetaMorph (Molecular Device) software. Images were further processed with imageJ.

## Generation of guinea pig anti-Hyx antibodies

The cDNA region encoding the N-terminal 1–176 amino acid residues of Hyx/Cdc73 was amplified by PCR with the oligos: 5′-TCCGAATTCATGGCAGATCCGCTCA GC-3′ and 5′-ATGCGGCCGCCTACGTCTCGGACAGCGACTT-3′. The PCR products were cloned into the pMAL-c2x (Addgene # 75286) vector. The fusion protein MBP-Hyx 1–176 was purified and injected into guinea pigs, and the antibodies generated were purified by GenScript (Hong Kong).

## Clonal analysis

MARCM clones were generated as previously described [54]. Briefly, larvae were heat shocked at 37˚C for 90 min at 24 h ALH and 10 to 16 h after the first heat shock. Larvae were further aged for 3 d at 25˚C, and larval brains were dissected and processed for immunohistochemistry. To generate type II NSC clones, UAS lines were crossed to the type II driver (*worniu* (*wor*)-Gal4, *ase*-Gal80^ts; *UAS-CD8-GFP*) at 25˚C and shifted to 29˚C at 24 h ALH. Wandering third instar larvae were dissected after incubation for 3 or 4 d at 29˚C. Z-stacks were acquired and NSC number per clone/lineage was manually counted. Percentage of ectopic NSCs refers to the percentage of NSC clones or lineages with ectopic NSCs out of total number of clones/lineages scored in this study. All clones in the larval brains scored in this study were NSC clones.

## Time-lapse recording

The time-lapse recording was performed as described [44]. The whole-mount brain expressing G147-GFP was used to analyze the asymmetric cell division of NSCs. The brain was dissected and loaded into a Lab-Tek chambered coverglass (Thermo Fisher Scientific) filled with dissecting medium that is supplemented with 2.5% methyl cellulose (Sigma-Aldrich). The time-lapse images of NSC divisions were acquired every 30 s on a confocal microscope (LSM 710; ZEISS). The video was processed with ImageJ and displayed at 15 frames per second.

## Microtubule regrowth assay

The microtubule regrowth assay was performed as described previously [44]. Third instar larval brains were dissected in Shield and Sang M3 insect medium (Sigma-Aldrich) supplemented with 10% FBS, and microtubules were depolymerized by incubating the larval brains on ice for 40 min. The brains were allowed to recover at 25˚C for various time periods to facilitate microtubule regrowth. The brains were immediately fixed in 10% formaldehyde in testis buffer (183 mM KCl, 47 mM NaCl, 10 mM Tris-HCl, and 1 mM EDTA (pH 6.8)) supplemented with 0.01% Triton X-100. The fixed brains were washed once in PBS and twice in 0.1% Triton X-100 in PBS, following which they were processed for immunohistochemistry. The mean immunofluorescence intensity of α-tub detected on astral microtubules proximal to

MTOC at interphase and spindle microtubules at metaphase were quantified on selected regions of the same size with ImageJ.

## S2 cell culture, transfection, and quantitative RT-PCR

**Cell culture.** *Drosophila* S2 cells were cultured in Express Five SFM (Thermo Fisher Scientific), supplemented with 2 mM glutamine (Thermo Fisher Scientific), at 25°C.

**Double-stranded RNA (dsRNA) production and interference.** DNA fragments, approximately 470 bp in length for ds-egfp as control and 825 bp in length for ds-hyx, were amplified using PCR. Each primer used in the PCR contained a 5′ T7 RNA polymerase binding site (TAATACGACTCACTATAGGG) followed by sequences specific for the targeted genes. The PCR products were purified by using the QIAquick PCR Purification Kit (Cat No. 28106). The purified PCR products were used as templates for the synthesis of dsRNA, by using a MEGA-SCRIPT T7 transcription kit (Ambion, Austin, TX). The dsRNA products were ethanol precipitated and resuspended in water. The dsRNAs were annealed by incubation at 65°C for 30 min followed by slow cooling to room temperature. To ensure that the majority of the dsRNA existed as a single band, 1 μg of dsRNA was analyzed by 1% agarose gel electrophoresis. S2 cells were cultured in 24-well plates at 50% to 90% confluency in 500 μl of medium. Cells were treated with 5 μg dsRNA and collected 72 h after transfection for mRNA extraction and immunostaining.

The primers used for dsRNA synthesis were the following:
ds-*egfp*-forward: 5′-TCGTGACCACCCTGACCTAC-3′;
s-*egfp*-reverse: 5′- GCTTCTCGTTGGGGTCTTT- 3′;
ds-*hyx*-forward: 5′-TGCTGCAACACTCGGTCTAC-3′;
ds-*hyx*-reverse: 5′-GTGCTCCCGGTAGGTTGTTA- 3′.

**Extraction of total messenger RNA (mRNA) and RT-qPCR.** Total mRNA was extracted from control (*UAS-β-Gal* RNAi; *UAS-β-Gal* RNAi) and *hyx* RNAi; *UAS-Dicer2* 48 h ALH instar larval brains under the control of *actin5C*-Gal4 driver using TRI Reagent (Sigma-Aldrich) according to the manufacturer's instructions. Reverse transcription was performed with iScript cDNA Synthesis Kit (Bio-RAD) according to the manufacturer's instructions. RT-qPCR was performed according to the manufacturer's instructions (SsoFast EvaGreen, Bio-RAD). References genes used as an internal control were as follows: *rp49/Rpl32* (*Ribosomal protein L32*), *Sdh* (*Succinate dehydrogenase*), and *Gapdh1* (*Glyceraldehyde 3 phosphate dehydrogenase 1*).

The primers pairs used for RT-qPCR were the following:
*hyx* forward: 5′-AGCCGGCTCGAATAGCCAAAC-3′;
*hyx* reverse: 5′-TGAGCATGGTAATGAGGCTTG-3′;
*γ-tub23C* forward: 5′-ACCGCAAGGATGTGTTCTTC-3′;
*γ-tub23C* reverse: 5′-CCTCCGTGCTTGGATAGGTA-3′;
*cnn* forward: 5′-CCGGCAGGATATCTAGCGTA-3′;
*cnn* reverse: 5′-TTGCTGTCCGGTGATGTAGA-3′;
*msps* forward: 5′-TTACGCGACCAAATGATGAC-3′;
*msps* reverse: 5′-TACACACCAGCGCCTTACTG-3′;
*tacc* forward: 5′-AGCACTTGCAAGCCATGAGT-3′;
*tacc* reverse: 5′-GCCTTCTGTTGATCCATGCT-3′;
*polo* forward: 5′-AGAGCCTGTACCAGCAGCTC-3′;
*polo* reverse: 5′-CTGCAGGATCTGTGTTCTCG-3′;
*aurA* forward: 5′-AAGAAGACCACATCAGAGTTTGC-3′;
*aurA* reverse: 5′-TTGATGTCCCTGTGTATGATGTC-3′;

*baz* forward: 5′-ATGACTGCTCATGGCAACAC-3′;
*baz* reverse: 5′-TCTGCGATGGATTAGCACTG-3′;
*par6* forward: 5′-AGCTGACCAACATCCAGTTTCT-3′;
*par6* reverse: 5′-CCATTTACCTCGATCACCTCAT-3′;
*aPKC* forward: 5′-ATGACCCACTTGGATTACGC-3′;
*aPKC* reverse: 5′-GCCGACTGAATGGAACTGAC-3′;
*insc* forward: 5′-TCTTCCGGCTGATTGATACC-3′;
*insc* reverse: 5′-TTGGTACACGGACGTGATGA-3′;
*pins* forward: 5′-CGGAAATCAGTCGGATGG-3′;
*pins* reverse: 5′-CCTGTGCTCGTAGCTTTTCC-3′;
*rp49* forward: 5′-TGTCCTTCCAGCTTCAAGATGACCATC-3′;
*rp49* reverse: 5′-CTTGGGCTTGCGCCATTTGTG-3′;
*Gapdh1* forward: 5′-ATGACGAAATCAAGGCTAAG-3′;
*Gapdh1* reverse: 5′-GAGTAACCGAACTCGTTGTC-3′;
*sdh* forward: 5′-GTCTGAAGATGCAGAAGACC-3′;
*sdh* reverse: 5′-ACAATAGTCATCTGGGCATT-3′.

## Chromatin immunoprecipitation

ChIP was performed according to the manufacturer's protocol (Cell Signaling, #9005). Sonicated lysates were used for ChIP with antibodies against Hyx (final bleed, J. T. Lis) and preimmune serum as a control. Immunoprecipitated DNA was analyzed by quantitative real-time PCR using specific primers to the potential promoter region of various centrosomal genes and negative control—the intergenic sequence at 5 kb downstream of *numb* genome as well as the positive control targeting the potential *orb2* promoter region:

*polo Pro* forward: 5′-TACCAGAAAGTGTGCGATAGCC-3′;
*polo Pro* reverse: 5′-GAAACGGAGATCAGATCCACAC-3′;
*polo Pro* forward: 5′-CCTGCAATTACAAGGTGGCA-3′;
*polo Pro* reverse: 5′-ACTAAACAGTCAACGGTCAACT-3′;
*aurA Pro* forward: 5′-TCGGCATCATATCATAAACGAC-3′;
*aurA Pro* reverse: 5′-TTATCGGGCATCTCTGAACA-3′;
*numb Pro* forward: 5′-CAGCCCAACAAGCCAATAAA-3′;
*numb Pro* reverse: 5′-GGGCGTGAGTAAATTGTCGT-3′;
negative control forward: 5′-TCCTTGGTCCTAACGTGGTC-3′;
negative control reverse: 5′-AAGTATTTGCCCCAGCTTGA-3′;
*orb2 Pro* forward: 5′-CTCCACAACGATTCCGATTT-3′;
*orb2 Pro* reverse: 5′-CCGCACCAACACTTTCTACA-3′.

## Luciferase assay

Luciferase assay was performed according to the manufacturer's protocol (Dual-Glo Luciferase Assay System (E2920), Promega). *Drosophila* S2 cells were cotransfected with 0.1 μg of *polo* promoter-luciferase reporter (*poloPro-luc*) and 0.1 μg of Venus-tagged *hyx*-FL. Luciferase reporter is made on the pGL3 basic reporter construct (Promega), and 643 bp sequence before the transcriptional start site of *polo* genome was cloned into the pGL3-basic vector bone. The Venus-tagged *hyx* is made by cloning full-length wild-type *hyx* into pAVW (*Drosophila* Genomics Resource Center [DGRC]). For negative control, pGL3-basic and empty pAVW was cotransfected into S2 cells. For normalizing transfection efficiency, 0.1 μg of plasmid encoding the *Renilla* gene was transfected. S2 cells were treated with *ds-egfp* and *ds-hyx* for 24 h before transfection with 0.1 μg of pGL3-*poloPro* and 0.1 μg of pRL-SV40 plasmids and cells were

harvested 48 h after transfection. For the internal control group, 0.1 μg of *actin5c* promoter containing pGL3 (*actin5c-luc*) was cotransfected with 0.1 μg of pRL-SV40 plasmid into S2 cells.

## Western blotting

Embryos were treated with 50% bleach for 2 min to dissolve chorion. After that, embryos or larval brains were homogenized in RIPA buffer (50 mM Tris HCl (pH 7.5), 150 mM NaCl, 1 mM EDTA, 1% Triton X-100, 0.5% sodium deoxycholate,0.1% SDS), and western blotting was carried out according to standard procedures.

## Quantification and statistical analysis

*Drosophila* larval brains from various genotypes were placed dorsal side up on confocal slides. The confocal z-stacks were taken from the surface to the deep layers of the larval brains. For each genotype, at least 10 NSCs were imaged and ImageJ or Zen software was used for quantifications.

The localization of polarity proteins was scored by 3 categories: "Strong crescent," "Weak crescent," and "No crescent."

Statistical analysis was essentially performed using GraphPad Prism 9. Unpaired two-tail *t* tests were used for comparison of 2 sample groups, and one-way ANOVA or two-way ANOVA followed by Sidak's multiple comparisons test was used for comparison of more than 2 sample groups. All data are shown as the mean ± SD. Statistically nonsignificant (ns) denotes $p > 0.05$, * denotes $p < 0.05$, ** denotes $p < 0.01$, *** denotes $p < 0.001$, and **** denotes $p < 0.0001$. All experiments were performed with a minimum of 2 repeats. In general, n refers to the number of NSCs counted unless otherwise indicated.

## Supporting information

**S1 Fig. Hyx regulates NSC homeostasis of *Drosophila* larval central brains.** (**A**) MARCM clones of control (FRT82B; $n = 20$), $hyx^{HT622}$ ($n = 23$), and $hyx^{W12-46}$ ($n = 25$) were labelled for Hyx, Dpn, and GFP. (**B**) Immunofluorescence intensity (with SD) of both nuclear and cytoplasmic in wild-type NSCs. Nuclear Hyx: 0.74 ± 0.09-fold; cytoplasmic Hyx: 0.26 ± 0.09-fold, $n = 15$ NSC. (**C**) Fold change of immunofluorescence intensity (with SD) of Hyx in NSCs from control (FRT82B), $hyx^{HT622}$, and $hyx^{W12-46}$ ($n = 40$). Control: 1 ± 0.23-fold, $n = 20$; $hyx^{HT622}$, 0.15 ± 0.18-fold, $n = 25$; $hyx^{W12-46}$, 0.32 ± 0.20-fold, $n = 28$. (**D**) Western blotting analysis of 24 h AEL embryo extracts of control, $hyx^{HT622}$, and $hyx^{W12-46}$ as well as third instar larval brain extracts of control (*UAS-β-gal* Ri) and *hyx* Ri; $hyx^{HT622/+}$ driven by *insc*-Gal4. Blots were probed with anti-Hyx antibody and anti-GAPDH antibody. A protein ladder was indicated on the left. (**E**) Fold change of Hyx protein levels normalizing against GAPDH (with SD) in D. Control (FRT82B): 1-fold; $hyx^{HT622}$, 0.31 ± 0.06-fold; $hyx^{W12-46}$, 0.26 ± 0.04-fold; control (*UAS-β-gal* Ri): 1-fold; *hyx* Ri; $hyx^{HT622/+}$: 0.22 ± 0.007-fold. Minimum 2 biological replicates for all blots. (**F**) Type I and type II NSC lineages of control (*UAS-β-Gal* RNAi), *hyx* RNAi (GD/V28318), and *hyx* RNAi (KK/V103555) under the control of *insc*-Gal4 driver were labeled for Dpn, Ase, and CD8-GFP. (**G**) Percentage of NSC lineages with multiple NSCs ($\geq 2$ NSCs) for genotypes in F. Type I: control (*UAS-β-Gal* RNAi), 0, $n = 57$; *hyx* RNAi (GD), 43.0%, $n = 92$; *hyx* RNAi (KK), 62.7%, $n = 73$. Type II: control (*UAS-β-Gal* RNAi), 0, $n = 64$; *hyx* RNAi (GD), 72.0%, $n = 83$; *hyx* RNAi (KK), 89.3%, $n = 56$. (H) Average NSC number per NSC lineage (with SD) for genotypes in F. Type I: control (*UAS-β-Gal* RNAi), 1.0, $n = 15$; *hyx* RNAi (GD), 1.6 ± 0.68, $n = 29$; *hyx* RNAi (KK), 2.5 ± 1.50, $n = 35$. Type II: control (*UAS-β-Gal* RNAi), 1.0, $n = 12$; *hyx* RNAi (GD), 3.4 ± 1.92, $n = 23$; *hyx* RNAi (KK), 6.2 ± 2.8, $n = 21$.

Statistical significances were determined by unpaired two-tailed Student $t$ test in B. One-way ANOVA with multiple comparison was performed in C, E, and H. ****$p < 0.0001$ in B, C, E, and H. In H, ns = 0.6846, **$p = 0.0086$, ***$p = 0.0002$. Clones are outlined with white dotted lines. NSCs and NSC-like cells are pointed by arrows. Scale bars: 5 μm. The underlying data for this figure can be found in the S1 Data. AEL, after egg laying; Hyx, Hyrax; MARCM, mosaic analysis with a repressible cell marker; NSC, neural stem cell; RNAi, RNA interference. (TIF)

**S2 Fig. Human Parafibromin/HRPT2 fully substitutes for Hyx in larval brains.** (**A**) Type I MARCM clones of control (FRT82B; $n = 20$), $hyx^{HT622}$ ($n = 17$), $hyx^{W12-46}$ ($n = 30$), $UAS\text{-}hyx$ $hyx^{HT622}$ ($n = 21$), and $UAS\text{-}hyx$ $hyx^{W12-46}$ ($n = 40$) were labelled for Dpn, Ase, and CD8. Ectopic NSCs were observed in 88.2% of $hyx^{HT622}$ and 36.7% of $hyx^{w12-46}$ larvae, but not in the control or rescued larvae. (**B**) Type II MARCM clones of control (FRT82B), $hyx^{HT622}$, $hyx^{W12-46}$, $UAS\text{-}hyx$ $hyx^{HT622}$, and $UAS\text{-}hyx$ $hyx^{W12-46}$ were labelled for Dpn, Ase, and CD8. Ectopic NSCs were observed in $hyx^{HT622}$ (81.0%, $n = 21$) and $hyx^{w12-46}$ (78.5%, $n = 30$) larvae, but not in control ($n = 20$), $UAS\text{-}hyx$ $hyx^{HT622}$ ($n = 17$) and $UAS\text{-}hyx$ $hyx^{W12-46}$ ($n = 40$) larvae. (**C**) MARCM clones of $UAS\text{-}HRPT2$ $hyx^{HT622}$ type I ($n = 30$) and type II ($n = 7$) were labelled for Dpn, Ase, and GFP. (**D**) Type I and type II NSC lineages from $hyx$ RNAi (V103555 with $UAS\text{-}CD8\text{-}GFP$) and $UAS\text{-}HRPT2; hyx$ RNAi under the control of $insc$-Gal4 driver were labelled for Dpn, Ase, and CD8 ($n = 10$ brain lobes for each genotype). (**E**) Type I ($n = 12$) and type II ($n = 16$) MARCM clones from $ctr9^{12P023}$ were labelled for Dpn, Ase, and CD8. Clones/lineages are outlined by white-dotted lines. NSCs are indicated by white arrows. Scale bars: 5 μm. $hyx$, Hyrax; MARCM, mosaic analysis with a repressible cell marker; NSC, neural stem cell; RNAi, RNA interference. (TIF)

**S3 Fig. Hyx governs asymmetric cell division of NSCs.** (**A**) INP lineages of control ($UAS\text{-}Dicer2$) and $hyx$ RNAi (KK/V103555 with $UAS\text{-}Dicer2$) driven by $erm$-Gal4, $UAS\text{-}CD8\text{-}GFP$ were labelled for Dpn, Ase, and CD8 ($n = 30$ for both). (**B**) INP lineages of control ($UAS\text{-}Dicer2$) and $hyx$ RNAi (KK with $UAS\text{-}Dicer2$) driven by $erm$-Gal4, $UAS\text{-}CD8\text{-}GFP$ were labelled for Hyx, Dpn, and CD8 ($n = 30$ and $n = 45$, respectively). (**C**) Metaphase NSCs of control ($insc{>}CD8\text{-}GFP$; $n = 50$) and $hyx$ RNAi (GD/V28318 with $UAS\text{-}CD8\text{-}GFP$) type II lineages were labeled for aPKC, Mira, CD8, and DNA. $hyx$ RNAi: aPKC delocalization, 100%, $n = 50$; Mira delocalization, 70%, $n = 50$. (**D**) Metaphase NSCs from control ($insc{>}CD8\text{-}GFP$) and $hyx$ RNAi (KK/V103555 with $UAS\text{-}CD8\text{-}GFP$) type II lineages were labeled for aPKC, Mira, CD8, and DNA. In $hyx$ RNAi, delocalization of aPKC: 100%; Mira: 90%; $n = 50$ for all. (**E**) Metaphase NSCs from control ($insc{>}CD8\text{-}GFP$) and $hyx$ RNAi (GD/V28318 with $UAS\text{-}CD8\text{-}GFP$) type II lineages were labeled with Insc, Numb, CD8, and DNA. $hyx$ RNAi, 100% delocalization of Insc and Numb; $n = 50$ for all. (**F**) Metaphase NSCs from control ($insc{>}CD8\text{-}GFP$) and $hyx$ RNAi (GD/V28318 with $UAS\text{-}CD8\text{-}GFP$) type II lineages were labeled for Baz, CD8, and DNA; $hyx$ RNAi: 100% delocalization of Baz; $n = 50$ for both genotypes. (**G**) Metaphase NSCs from control ($insc{>}CD8\text{-}GFP$) and $hyx$ RNAi (GD/V28318 with $UAS\text{-}CD8\text{-}GFP$) type II lineages were labeled for Par6, CD8, and DNA. $hyx$ RNAi: Par6 delocalization, 100%; $n = 50$ for both genotypes. (**H**) Metaphase NSCs from control ($insc{>}CD8\text{-}GFP$) and $hyx$ RNAi (GD/V28318 with $UAS\text{-}CD8\text{-}GFP$) type II lineages were labeled for Pins, CD8, and DNA. $hyx$ RNAi: Pins delocalization, 100%; $n = 50$ for both genotypes. (**I**) Metaphase NSCs from control ($insc{>}CD8\text{-}GFP$; $n = 50$) and $hyx$ RNAi (GD/V28318 with $UAS\text{-}CD8\text{-}GFP$) type II lineages were labeled for Brat, CD8, and DNA. $hyx$ RNAi: Brat delocalization, 100%; $n = 50$ for both genotypes. INP lineages are outlined by white-dotted lines. Scale bars: 5 μm. aPKC, atypical PKC; Baz, Bazooka; Hyx, Hyrax; Insc, Inscuteable; MARCM, mosaic analysis with a repressible

cell marker; Mira, Miranda; NSC, neural stem cell; RNAi, RNA interference.
(TIF)

**S4 Fig. Hyx does not influence PI3K signaling or cell cycle regulators, including PP2A and Stg.** (**A**) NSC lineages from control (*UAS-β-Gal* RNAi, *n* = 54) and *hyx* RNAi (KK, *n* = 55) were labelled for P-Akt (Ser505), Dpn, and GFP. (**B**) Fold change of the immunofluorescence intensity of p-Akt (with SD) in NSCs for genotypes in A. Control: 1-fold; *hyx* RNAi: 1.02 ± 0.05-fold. (**C**) NSC lineages from control (*UAS-β-Gal* RNAi, *n* = 46) and *hyx* RNAi (KK, *n* = 65) were stained for Stg, Dpn, and GFP. (**D**) Fold change of the immunofluorescence intensity of Stg (with SD) in NSCs for genotypes in C. Control: 1-fold; *hyx* RNAi: 1.01 ± 0.06-fold. (**E**) NSCs from MARCM clones of control (FRT82B; *n* = 11) and $hyx^{HT622}$ (*n* = 15) were probed with Mts, Dpn, GFP, and DNA. (**F**) Immunofluorescence intensity of Mts after normalization against Dpn (with SD) for genotypes in E. Control: 0.89 ± 0.39-fold; $hyx^{HT622}$: 0.78 ± 0.64-fold. RNAi was controlled by *insc*-Gal4 in A and C. White arrows indicate NSCs in A. NSC lineages/NSCs are outlined by white-dotted lines in C and E. Statistical significances were determined by unpaired two-tailed Student *t* test in B, D, and F. ns = 0.6846 in B; ns = 0.8036 in D; ns = 0.6358 in F. Scale bars: 5 μm. The underlying data for this figure can be found in the S1 Data. Hyx, Hyrax; MARCM, mosaic analysis with a repressible cell marker; Mts, microtubule star; NSC, neural stem cell; PP2A, phosphatase 2A; RNAi, RNA interference; Stg, String.
(TIF)

**S5 Fig. Hyx is required for the formation of microtubule aster and centriole number in NSCs.** (**A**) Interphase NSCs from control (*UAS-β-Gal* RNAi; 100% aster formation, *n* = 20) and *hyx* RNAi (KK/V103555; *n* = 22) were labelled for α-tub, Asl, and PH3. (**B**) Interphase NSCs of control (*UAS-β-Gal* RNAi; *n* = 23) and *hyx* RNAi (KK/V103555; *n* = 20) were labelled for Sas-4, Asl, and PH3. (**C**) Prometa/metaphase NSCs of control (*UAS-β-Gal* RNAi) and *hyx* RNAi (KK/V103555) were labeled for Sas-4, Asl, and PH3 (*n* = 23 for both). (**D**) Interphase and metaphase NSCs from control and $hyx^{HT622}$ MARCM clones were probed with Asl, GFP, DNA, and PH3. Centrioles marked by Asl are pointed out by arrows. Control interphase (*n* = 21) and metaphase NSCs (*n* = 20) typically contain 2 Asl-positive centrioles. Two centrioles marked by Asl were always seen in $hyx^{HT622}$ interphase NSCs (*n* = 20); multiple centrioles labelled by Asl were observed in 28.1% (*n* = 32) metaphase NSCs from $hyx^{HT622}$ MARCM clones and the rest of metaphase (79.1%) NSCs showing 2 Asl-positive centrioles. NSCs/NSC lineages are outlined and Zoom-in areas are boxed. (**E**) $hyx^{HT622}$ MARCM clones were labelled with GFP, Dpn, and Ase. Cytokinesis delay was shown (*n* = 23). (**F**) Metaphase ds-*egfp*-treated S2 cells (*n* = 195) and ds-*hyx*-treated S2 cells (*n* = 172) were labeled for Msps and Asl. (**G**) Quantification graph displaying the percentage of metaphase S2 cells with the indicated number of Asl per NSC in G. Percentage of metaphase S2s with multiple Asl (≥3): ds-*egfp*, 52.0 ± 7.3%; ds-*egfp*, 25.1 ± 15.9%. *hyx* knockdown was driven by *insc*-Gal4 in A-C. Arrows indicate the centrosomes. Scale bars: 5 μm. The underlying data for this figure can be found in the S1 Data. Asl, Asterless; Hyx, Hyrax; MARCM, mosaic analysis with a repressible cell marker; Msps, Mini spindles; NSC, neural stem cell; RNAi, RNA interference; Sas-4, Spindle assembly abnormal 4.
(TIF)

**S6 Fig. Hyx is required for the recruitment of PCM proteins γ-tub and Cnn to the centrosome in NSCs.** (**A**) Interphase NSCs of MARCM clones in control (FRT82B) and $hyx^{HT622}$ were labelled for γ-tub, Asl, GFP, and PH3. γ-tub delocalization: control, 0%, *n* = 26; $hyx^{HT622}$, 95.7%, *n* = 23; $hyx^{w12-46}$, 54.8%, *n* = 42. (**B**) Metaphase NSCs of MARCM clones in control

(FRT82B) and *hyx*^*HT622*^ were labelled for γ-tub, Asl, GFP, and PH3. γ-tub delocalization: control, 0%, *n* = 29; *hyx*^*HT622*^, 93.1%, *n* = 29; *hyx*^*w12-46*^, 70.8%, *n* = 24. (**C**) Quantification graph of the fold change of γ-tub intensity (with SD) in NSCs from A and B. Interphase: control, 1-fold; *hyx*^*HT622*^, 0.15 ± 0.04-fold; *hyx*^*w12-46*^, 0.35 ± 0.17-fold. Metaphase: control, 1-fold; *hyx*^*HT622*^, 0.17 ± 0.12-fold; *hyx*^*w12-46*^, 0.53 ± 0.04-fold. (**D**) Control (*UAS-β-Gal* RNAi; *n* = 36) and *hyx* RNAi (KK/V103555; *n* = 42) interphase NSCs were labelled for γ-tub, Asl, GFP, and PH3. (**E**) Control (*UAS-β-Gal* RNAi; *n* = 25) and *hyx* RNAi (KK/V103555; *n* = 32) metaphase NSCs were labelled for γ-tub, Asl, GFP, and PH3. In control, robust distribution of γ-tub was seen in 88.9% of interphase (A) and 92.0% of metaphase (B) NSCs. (**F**) Quantification graph of the fold change of γ-tub intensity (with SD) in NSCs from D and E. Interphase: control, 1-fold, *n* = 36; *hyx* RNAi, 0.32 ± 0.13-fold, *n* = 42. Metaphase: control, 1-fold, *n* = 25; *hyx* RNAi, 0.38 ± 0.03-fold, *n* = 32. (**G**) Interphase NSCs of control (FRT82B; *n* = 51) and *hyx*^*HT622*^ (*n* = 33) MARCM clones were labelled for Cnn, Asl, GFP, and PH3. (**H**) Metaphase MARCM clones of control (FRT82B; all NSCs have robust Cnn localization, *n* = 27) and *hyx*^*HT622*^ (*n* = 29) were labelled for Cnn, Asl, GFP, and PH3. (**I**) Quantification graph of the fold change of Cnn intensity (with SD) in NSCs from G and H. Interphase: control, 1-fold, *n* = 51; *hyx*^*HT622*^, 0.09 ± 0.06-fold, *n* = 33; *hyx*^*w12-46*^, 0.14 ± 0.03-fold, *n* = 12. Metaphase: control, 1-fold, *n* = 27; *hyx*^*HT622*^, 0.44 ± 0.02-fold, *n* = 29; *hyx*^*w12-46*^, 0.59 ± 0.23-fold, *n* = 18. (**J**) Interphase NSCs from control (*UAS-β-Gal* RNAi; 96.3% robust Cnn signal) and *hyx* RNAi (KK/V103555) were labelled for Cnn, Asl, GFP, and DNA (*n* = 27 for both). (**K**) Metaphase NSCs of control (*UAS-β-Gal* RNAi; 100% Cnn robust localization, *n* = 24) and *hyx* RNAi (KK/V103555) were labelled for Cnn, Asl, GFP, and DNA. (**L**) Quantification graph of the fold change of Cnn intensity (with SD) in NSCs from J and K. Interphase: control, 1-fold, *n* = 27; *hyx* RNAi, 0.13 ± 0.17-fold, *n* = 27. Metaphase: control, 1-fold, *n* = 24; *hyx* RNAi, 0.74 ± 0.28-fold, *n* = 30. *hyx* knockdown was driven by *insc*-Gal4 in A-I. NSCs are circled by dotted lines. Arrows indicate the centrosomes. Statistical significances were determined by two-way ANOVA with multiple comparison in C, F, and L. One-way ANOVA was performed in I. In C, \*\*\*\**p* < 0.0001, \*\*\**p* = 0.0002; in F, \*\*\**p* = 0.0010, \*\**p* = 0.0014; in I, \*\*\**p* = 0.0002, \*\**p* = 0.0022, \**p* = 0.0106; in L, \*\*\*\**p* < 0.0001, \*\**p* = 0.0015. Scale bars: 5 μm. The underlying data for this figure can be found in the S1 Data. Asl, Asterless; Cnn, centrosomin; Hyx, Hyrax; MARCM, mosaic analysis with a repressible cell marker; NSC, neural stem cell; PCM, pericentriolar material; RNAi, RNA interference; γ-tub, γ-tubulin.
(TIF)

**S7 Fig. Msps and D-TACC were delocalized from the centrosomes in *hyx*-deficient NSCs.**
(**A**) Interphase NSCs of MARCM clones in control (FRT82B; 98.1% of NSCs have strong Msps localization at the centrosomes) and *hyx*^*HT622*^ were analyzed by Msps, Asl, GFP, and PH3. Control, *n* = 54; *hyx*^*HT622*^, *n* = 44; *hyx*^*w12-46*^, *n* = 17. (**B**) Metaphase NSCs of control (FRT82B; all NSCs have Msps localization at the centrosomes, *n* = 32), *hyx*^*HT622*^ (*n* = 24), and *hyx*^*w12-46*^ (*n* = 15) MARCM clones were labelled for Msps, Asl, GFP, and PH3. (**C**) Quantification graph of the fold change of Msps intensity (with SD) in NSCs from A and B. Interphase: control, 1-fold, *n* = 54; *hyx*^*HT622*^, 0.14 ± 0.04-fold, *n* = 44; *hyx*^*w12-46*^, 0.64 ± 0.04-fold, *n* = 17. Metaphase: control, 1-fold, *n* = 40; *hyx*^*HT622*^, 0.19 ± 0.07-fold, *n* = 24; *hyx*^*w12-46*^, 0.62 ± 0.20-fold, *n* = 15. (**D**) Interphase NSCs of control (*UAS-β-Gal* RNAi) and *hyx* RNAi (KK/V103555) were labelled for Msps, Asl, and PH3. Msps delocalization at the centrosomes: control: 0%, *n* = 55; *hyx* RNAi, 82.5%, *n* = 57. (**E**) Metaphase NSCs of control (*UAS-β-Gal* RNAi) and *hyx* RNAi (KK/V103555) were labelled for Msps, Asl, and PH3. Msps localization at the centrosomes: control: 98.5%, *n* = 55; *hyx* RNAi, 16.4%, *n* = 67. (**F**) Quantification graph of the fold change of Msps intensity (with SD) in NSCs from D and E. Interphase: control, 1-fold, *n* = 25; *hyx* RNAi,

0.07 ± 0.03-fold, $n = 25$. Metaphase: control, 1-fold, $n = 25$; *hyx* RNAi, 0.10 ± 0.11-fold, $n = 26$. (**G**) Interphase NSCs from control (FRT82B; $n = 41$) and $hyx^{HT622}$ ($n = 25$) MARCM clones were labelled for DTACC, Asl, GFP, and PH3. (**H**) MARCM clones of control (FRT82B; 100% D-TACC localization, $n = 18$) and $hyx^{HT622}$ ($n = 21$) were labelled for DTACC, Asl, GFP, and PH3. (**I**) Quantification graph of the fold change of DTACC intensity (with SD) in NSCs from G and H. Interphase: control, 1-fold, $n = 21$; $hyx^{HT622}$, 0.13 ± 0.01-fold, $n = 25$; $hyx^{w12-46}$, 0.22 ± 0.06-fold, $n = 22$. Metaphase: control, 1-fold, $n = 18$; $hyx^{HT622}$, 0.06 ± 0.007, $n = 21$; $hyx^{w12-46}$, 0.15- ± 0.17-fold, $n = 15$. (**J**) Interphase NSCs of control (*UAS-β-Gal* RNAi) and *hyx* RNAi (KK/V103555) were labelled for DTACC, Asl, and PH3. DTACC delocalization at the centrosomes: control, 3.8%, $n = 26$; *hyx* RNAi, 83.3%, $n = 24$. (**K**) Metaphase NSCs of control (*UAS-β-Gal* RNAi) and *hyx* RNAi (KK/V103555) were labelled for DTACC, Asl, and PH3. DTACC delocalization at the centrosomes: control, 3.3%, $n = 30$; *hyx* RNAi, 85.3%, $n = 34$. (**L**) Quantification graph of the fold change of DTACC intensity (with SD) in NSCs from J and K. Interphase: control, 1-fold, $n = 26$; *hyx* RNAi, 0.13 ± 0.05-fold, $n = 24$. Metaphase: control, 1-fold, $n = 30$; *hyx* RNAi, 0.20 ± 0.16-fold, $n = 34$. *hyx* knockdown was driven by *insc*-Gal4 in D and E and J and K. NSCs are circled by white-dotted lines. Centrosomes are pointed by arrows. Statistical significances were determined by two-way ANOVA with multiple comparison in C, F, I, and L. ****$p < 0.0001$ in C, I, and L; in C, ***$p = 0.0006$ for interphase and ***$p = 0.0005$ for metaphase; in F, ***$p = 0.0002$ for both. Scale bars: 5 μm. The underlying data for this figure can be found in the S1 Data. Asl, Asterless; Hyx, Hyrax; MARCM, mosaic analysis with a repressible cell marker; Msps, Mini spindles; NSC, neural stem cell; RNAi, RNA interference.
(TIF)

**S8 Fig. Polo and AurA were delocalized from the centrosomes upon *hyx* knockdown in NSCs and S2 cells.** (**A**) Interphase NSCs of control (*UAS-β-Gal* RNAi; $n = 29$) and *hyx* RNAi (KK/V103555; $n = 26$) were labelled with Polo, Asl, and PH3. (**B**) Metaphase NSCs of control (*UAS-β-Gal* RNAi; Polo present at the centrosomes in 95.0% of NSCs, $n = 20$) and *hyx* RNAi (KK/V103555; $n = 23$) were labelled for Polo, Asl, and DNA. (**C**) Quantification graph showing the fold change of Polo intensity (with SD) in A and B. Interphase: control, 1-fold, $n = 29$; *hyx* RNAi, 0.12 ± 0.06-fold, $n = 26$. Metaphase: control, 1-fold, $n = 20$; *hyx* RNAi, 0.26 ± 0.18-fold, $n = 24$. (**D**) Interphase NSCs of control (*UAS-β-Gal* RNAi; AurA present at the centrosomes in 96.6% of NSCs, $n = 29$) and *hyx* RNAi (KK/V103555; $n = 24$) were labelled for AurA, Asl, GFP, and PH3. (**E**) Metaphase NSCs of control (*UAS-β-Gal* RNAi; AurA observed at the centrosomes in 96.4% of NSCs, $n = 28$) and *hyx* RNAi (KK/V103555; $n = 30$) were labelled for AurA, Asl, GFP, and PH3. Polo and AurA are properly localized in all control NSCs in A-E. (**F**) Quantification graph showing the fold change of AurA intensity (with SD) in D and E. Interphase: control, 1-fold, $n = 29$; *hyx* RNAi, 0.04 ± 0.02-fold, $n = 24$. Metaphase: control, 1-fold, $n = 28$; *hyx* RNAi, 0.53 ± 0.08-fold, $n = 30$. (**G**) Metaphase cells from ds-*egfp*-treated S2 cells and ds-*hyx*-treated S2 cells were labelled for Ana2, γ-tub, and DNA. (**H**) Quantification graph showing the fold change of γ-tub intensity in G. ds-*egfp*, 1-fold, $n = 69$; ds-*hyx*, 1.03 ± 0.06-fold, $n = 52$. *hyx* knockdown was under the control of *insc*-Gal4 in A-F. NSCs are outlined by white-dotted lines. Centrosomes are pointed by arrows. Statistical significances were determined by two-way ANOVA with multiple comparison in C and F. Unpaired two-tailed Student $t$ test was performed in H. In C, **$p = 0.0015$ for interphase, **$p = 0.0029$ for metaphase; in F, ****$p < 0.0001$, ***$p = 0.0006$; in H, ns = 0.4023. Scale bars: 5 μm. The underlying data for this figure can be found in the S1 Data. Ana2, Anastral spindle 2; Asl, Asterless; AurA, Aurora-A; Hyx, Hyrax; NSC, neural stem cell; RNAi, RNA interference; γ-tub, γ-

tubulin.
(TIF)

**S9 Fig. Hyx is required for NSC polarity and centrosome assembly in early larval stages.**
(**A**) 24 h ALH brains from control (*UAS-β-Gal* RNAi; *n* = 10 brain lobes) and *hyx* RNAi
*hyx^HT622^* (KK/V103555, *n* = 10 brain lobes) were labelled with Hyx, Dpn, and Mira. Arrows
indicate NSCs. (**B**) 24 h ALH brains from control (*UAS-β-Gal* RNAi; both aPKC and Mira
formed crescent in all metaphase NSCs, *n* = 47) and *hyx* RNAi *hyx^HT622/+^* (aPKC and Mira
delocalized in 96.8% and 91.9% of NSCs, *n* = 62 for both) were examined by aPKC, Mira, and
PH3. (**C**) At 24 h ALH, interphase NSCs of control (*UAS-β-Gal* RNAi; γ-tub present at the
centrosomes in all NSCs examined, *n* = 34) and *hyx* RNAi *hyx^HT622^* (KK/V103555; γ-tub delo-
calization at the centrosomes, 82.4%, *n* = 34) were labelled with γ-tub, Asl, aPKC, and PH3.
(**D**) At 24 h ALH, metaphase NSCs of control (*UAS-β-Gal* RNAi; γ-tub present at the centro-
somes in all NSCs observed, *n* = 34) and *hyx* RNAi *hyx^HT622^* (KK/V103555; γ-tub delocaliza-
tion at the centrosomes, 85%, *n* = 40) were labelled for γ-tub, Asl, aPKC, and PH3. (**E**)
Quantification graph showing the fold change of γ-tub intensity (with SD) in C and D. Inter-
phase: control, 1-fold, *n* = 34; *hyx* RNAi *hyx^HT622/+^*, 0.31 ± 0.04-fold, *n* = 34. Metaphase: con-
trol, 1-fold, *n* = 34; *hyx* RNAi *hyx^HT622/+^*, 0.31 ± 0.04-fold, *n* = 40. (**F**) Interphase NSCs of 24 h
ALH control (*UAS-β-Gal* RNAi; Polo present at the centrosomes in 96.4% of NSCs, *n* = 28)
and *hyx* RNAi *hyx^HT622^* (KK/V103555; Polo delocalization at the centrosomes, 81.8%, *n* = 22)
were labelled for Polo, Asl, aPKC, and PH3. (**G**) Metaphase NSCs of 24 h ALH control (*UAS-
β-Gal* RNAi; Polo observed at the centrosomes in 95.5% of NSCs, *n* = 52) and *hyx* RNAi
*hyx^HT622/+^* (KK/V103555; Polo mislocalization at the centrosomes, 71.9%, *n* = 32) were labelled
for Polo, Asl, aPKC, and PH3. (**H**) Quantification graph showing the fold change of Polo
intensity (with SD) in F and G. Interphase: control, 1-fold, *n* = 28; *hyx* RNAi *hyx^HT622/+^*,
0.29 ± 0.17-fold, *n* = 22. Metaphase: control, 1-fold, *n* = 52; *hyx* RNAi *hyx^HT622/+^*,
0.36 ± 0.10-fold, *n* = 32. NSCs are outlined by white-dotted lines; centrosomes are indicated by
arrows; and aPKC and PH3 are probed in the same channel in C and D and F and G. Statistical
significances were determined by two-way ANOVA with multiple comparison in E and H. In
E, ****$p < 0.0001$ for both; in H, **$p = 0.0039$ for interphase, **$p = 0.0059$ for metaphase. Scale
bars: 5 μm. The underlying data for this figure can be found in the S1 Data. ALH, after larval
hatching; aPKC, atypical PKC; Asl, Asterless; Hyx, Hyrax; Mira, Miranda; NSC, neural stem
cell; RNAi, RNA interference; γ-tub, γ-tubulin.
(TIF)

**S10 Fig. Hyx is essential for NSC polarity and centrosome assembly in late larval stages.**
(**A**) Type I and type II NSC lineages from control (*UAS-β-Gal* RNAi; a single NSC was dis-
played in all lineages, type I, *n* = 20 and type II, *n* = 20) and *hyx* RNAi *hyx^HT622/+^* (KK/
V103555; ectopic NSCs were observed, type I, 84.3%, *n* = 51 and type II, 93.3%, *n* = 30) were
labelled with Dpn, Ase, and Mira. (**B**) NSC lineages from control (*UAS-β-Gal* RNAi; *n* = 50)
and *hyx* RNAi *hyx^HT622/+^* (KK/V103555; Hyx protein levels were reduced in the nucleus of
NSCs, 89.1%, *n* = 44) were examined with Hyx, Dpn, and Mira. (**C**) Interphase NSCs of con-
trol (*UAS-β-Gal* RNAi; γ-tub present at the centrosomes in all NSCs, *n* = 24) and *hyx* RNAi
*hyx^HT622/+^* (KK/V103555; Delocalization of γ-tub at the centrosomes in 88.9% of NSCs,
*n* = 19) were labelled with γ-tub, Asl, and PH3. (**D**) Metaphase NSCs of control (*UAS-β-Gal*
RNAi; γ-tub present at the centrosome in all NSCs, *n* = 25) and *hyx* RNAi *hyx^HT622/+^* (KK/
V103555; reduction of γ-tub protein levels at the centrosomes in 88.9% of NSCs, *n* = 18) were
labelled for γ-tub, Asl, and PH3. (**E**) Quantification graph showing the fold change of γ-tub
intensity (with SD) in C and D. Interphase: control, 1-fold, *n* = 24; *hyx* RNAi *hyx^HT622/+^*,
0.31 ± 0.05-fold, *n* = 19. Metaphase: control, 1-fold, *n* = 25; *hyx* RNAi *hyx^HT622/+^*,

0.29 ± 0.07-fold, $n$ = 18. (**F**) Interphase NSCs of control (*UAS-β-Gal* RNAi; Polo present at the centrosomes in all NSCs examined, $n$ = 27) and *hyx* RNAi *hyx*[HT622/+] (KK/V103555; Polo delocalized at the centrosome in 84.6% of NSCs, $n$ = 39) were labelled for Polo, Asl, aPKC, and PH3. (**G**) Metaphase NSCs of control (*UAS-β-Gal* RNAi; Polo observed at the centrosomes in all NSCs, $n$ = 27) and *hyx* RNAi *hyx*[HT622/+] (KK/V103555; Polo reduced at the centrosomes in 90.0% of NSCs, $n$ = 20) were labelled for Polo, Asl, aPKC, and PH3. (**H**) Quantification graph showing the fold change of Polo intensity (with SD) in F and G. Interphase: control, 1-fold, $n$ = 27; *hyx* RNAi *hyx*[HT622/+], 0.13 ± 0.03-fold, $n$ = 39. Metaphase: control, 1-fold, $n$ = 27; *hyx* RNAi *hyx*[HT622/+], 0.21 ± 0.10-fold, $n$ = 20. (**I**) S2 cells transfected with Venus-tagged pAFW (DGRC) and Venus-tagged *hyx*-FL were probed with Cas-3, GFP, and DNA. (**J**) Percentage of S2 cells (with SD) positive for GFP and Cas-3 for conditions in I. Venus-Vector: 11.2 ± 8.91%, $n$ = 392; Venus-hyx-FL; 54.3 ± 3.25%, $n$ = 397. (**K**) RT-qPCR analysis for various polarity genes in 48 h ALH larval brains from control (*UAS-β-gal* RNAi; *UAS-β-gal* RNAi) and *hyx* RNAi (KK) with *UAS-Dicer2* (*hyx* RNAi; *UAS-Dicer2*). After normalization against control (with SD): control, 1-fold; *hyx*, 0.23 ± 0.05-fold; *baz*, 0.84 ± 0.51-fold; *par6*, 0.69 ± 0.21-fold; *aPKC*, 5.21 ± 4.40-fold; *pins*, 1.04 ± 0.36-fold; *insc*, 1.07 ± 0.51-fold; *numb*, 1.8 ± 1.04-fold. Minimum of 3 repeats were conducted. NSC lineages are outlined by white-dotted lines in A and B. NSCs are circled by dotted lines. White arrows indicate NSCs (A, B) and centrosomes (C-G), respectively. aPKC and PH3 were probed in the same channel in F and G. Yellow arrows points at S2 cells double-positive for Cas-3 and GFP in I. (A, B) and centrosomes (C-G), respectively. Statistical significances were determined by two-way ANOVA with multiple comparison in E and H. Unpaired two-tailed Student $t$ test was performed in J and K. In E, ***$p$ = 0.0002 for both; in H, ***$p$ = 0.0001 for interphase, ***$p$ = 0.0002 for metaphase; in J, *$p$ = 0.0234; in K, ****$p$ < 0.0001 for *hyx*, ns = 0.5532 for *baz*, *$p$ = 0.0209 for *par6*, ns = 0.1043 for *aPKC*, ns = 0.8420 for *pins*, ns = 0.7802 for *insc*, ns = 0.1685 for *numb*. Minimum 3 biological replicates. Scale bars: 5 μm. The underlying data for this figure can be found in the S1 Data. aPKC, atypical PKC; Asl, Asterless; DGRC, Drosophila Genomics Resource Center; Hyx, Hyrax; hyx-FL, full-length hyx; Mira, Miranda; NSC, neural stem cell; RNAi, RNA interference; RT-qPCR, reverse transcription quantitative real-time PCR; γ-tub, γ-tubulin. (TIF)

**S1 Movie.** Time-lapse imaging of G147/+ (Jupiter-GFP) NSCs under the control of *insc*-Gal4 at 48 h ALH larval brains. Time scale: minute: second. Scale bar: 5 μm. (AVI)

**S2 Movie.** Time-lapse imaging of *hyx* RNAi (KK/V103555); G147 NSCs under the control of *insc*-Gal4 at 48 h ALH larval brains. Time scale: minute: second. Scale bar: 5 μm. (AVI)

**S1 Data.** Original quantification data for main figures and supporting information figures were provided. Mean, standard deviation, standard error of mean, mean difference, or standard error of difference was also indicated for each set of data analyzed by GraphPad Prism 9. (XLSX)

**S1 Raw Images. Original uncropped western blot for S1D Fig.** Hyx protein level is reduced in *hyx* loss of function larval brains. Western blotting analysis of 24 h AEL embryo extracts of control, *hyx*[HT622], and *hyx*[W12-46] as well as third instar larval brain extracts of control (*UAS-β-gal* Ri) and *hyx* Ri; *hyx*[HT622/+] driven by *insc*-Gal4. Blots were probed with anti-Hyx antibody (upper panels) and anti-GAPDH antibody (lower panels). Both "Unlabeled" (left panels) and "Labeled" (right panels) uncropped original blots were provided. A protein ladder was indicated on the left of the "Labeled" membrane. Cropped images shown in S1D Fig were boxed

and antibodies used were indicated by arrows. **Original uncropped western blot for Fig 6P. Hyx is dramatically decreased upon *hyx* knockdown under the control of *actin5C*-Gal4.** Western blotting analysis of larval brain protein extracts of control *(UAS-β-Gal* RNAi; *UAS-β-Gal* RNAi) and *hyx* knockdown with *UAS-Dicer2* (*hyx* RNAi; *UAS-Dicer2* RNAi) driven by *actin5C*-Gal4 at 48 h ALH. Blots were probed with anti-Hyx antibody (upper panels) and anti-GAPDH antibody (lower panels). Both "Unlabeled" (left panels) and "Labeled" (right panels) uncropped original blots were provided. A protein ladder was indicated on the left of the "Labeled" membrane. Cropped images used in Fig 6P were boxed and antibodies used were indicated by arrows. **Original uncropped western blot for Fig 6R. Polo and AurA protein levels were significantly decreased upon *hyx* knockdown under the control of *actin5C*-Gal4.** Western blotting analysis of 48 h ALH larval brain extracts of control *(UAS-β-Gal* RNAi; *UAS-β-Gal* RNAi) and *hyx* knockdown with *UAS-Dicer2* (*hyx* RNAi; *UAS-Dicer2* RNAi) under the control of *actin5C*-Gal4. Blots were probed with anti-Polo antibody (upper panels), anti-AurA antibody (middle panels), and anti-GAPDH antibody (lower panels). Both "Unlabeled" (left panels) and "Labeled" (right panels) uncropped original blots were provided. A protein ladder was indicated on the left of the "Labeled" membrane. Cropped images used in Fig 6R were boxed and antibodies used were indicated by arrows.
(PDF)

## Acknowledgments

We thank Dr. Yu F. for providing the EMS mutant collection (on 3R chromosome) from which *hyx* alleles were isolated. We also thank Konard Basler, J. A. Knoblich, M. Buszczak, J. T. Lis, F. Matsuzaki, W. Chia, X. Yang, J. Skeath, F. Yu, YN Jan, C. Gonzalez, J. Raff, E. Schejter, T. Megraw and C. Sunkel, the Bloomington Drosophila Stock Center, the Vienna Drosophila Resource Center, the Kyoto stock centre DGGR, and the Developmental Studies Hybridoma Bank for fly stocks and antibodies.

## Author Contributions

**Conceptualization:** Qiannan Deng, Cheng Wang, Hongyan Wang.

**Data curation:** Qiannan Deng, Cheng Wang, Chwee Tat Koe, Jan Peter Heinen, Ye Sing Tan, Song Li, Wing-Kin Sung.

**Formal analysis:** Qiannan Deng, Cheng Wang, Chwee Tat Koe, Jan Peter Heinen, Ye Sing Tan, Song Li, Wing-Kin Sung.

**Funding acquisition:** Cayetano Gonzalez, Hongyan Wang.

**Methodology:** Qiannan Deng, Cheng Wang, Chwee Tat Koe, Jan Peter Heinen, Ye Sing Tan, Song Li, Wing-Kin Sung.

**Resources:** Hongyan Wang.

**Supervision:** Cayetano Gonzalez, Hongyan Wang.

**Writing – original draft:** Qiannan Deng, Cheng Wang, Hongyan Wang.

**Writing – review & editing:** Qiannan Deng, Cayetano Gonzalez, Hongyan Wang.

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
