## [Editor Report · Decision Letter 0]

11 May 2022

Dear Dr Wang, 

Thank you for submitting a revised version of your manuscript entitled "Parafibromin/Hyrax Governs Cell Polarity and Centrosome Assembly in Drosophila Neural Stem Cells" for consideration as a Research Article by PLOS Biology.

Your manuscript has now been evaluated by the PLOS Biology editorial staff and I am writing to let you know that we would like to send your submission back to the reviewers for another look.

Once your full submission is complete, your paper will undergo a series of checks in preparation for peer review. Once your manuscript has passed the checks it will be sent out for review. To provide the metadata for your submission, please Login to Editorial Manager (https://www.editorialmanager.com/pbiology) within two working days, i.e. by May 13 2022 11:59PM.

Kind regards,

Lucas

Lucas Smith

Associate Editor

PLOS Biology

lsmith@plos.org

---

## [Decision Letter · Decision Letter 1]

17 Jun 2022

Dear Dr Wang,

Thank you for your patience while we considered your revised manuscript "Parafibromin/Hyrax Governs Cell Polarity and Centrosome Assembly in Drosophila Neural Stem Cells" for publication as a Research Article at PLOS Biology. Your revised study has been evaluated by the PLOS Biology editors, the Academic Editor and the original reviewers.

You can read the full reviews below my signature. As you will see all three reviewers agree that the manuscript has been strengthened, however, Reviewers 1 and 2 have a number of lingering concerns which we think should be addressed before publication. As a note, Reviewer 2 has commented that it would be necessary to perform additional rescue experiments. While we agree that these experiments would strengthen the study, after discussion with the Academic Editor, we think that these may be beyond the scope of a revision and we would not strictly require this new data for publication in PLOS Biology.

While we cannot accept the manuscript in its current form, we would welcome a revised manuscript that thoroughly addresses the remaining reviewer comments. We will then assess your revised manuscript and your response to the reviewers' comments with our Academic Editor aiming to avoid further rounds of peer-review, although might need to consult with the reviewers, depending on the nature of the revisions.

As some of the remaining reviewer requests may require generation of new data, we have granted a 3 month month deadline for your revision. Please email us (plosbiology@plos.org) if you have any questions or concerns, or would like to request an extension. 

**IMPORTANT - As you address the reviewer comments, please also make sure to address the following editorial requests:

1) FINANCIAL DISCLOSURES: Please update your financial disclosure statement to describe the role of any sponsors or funders in the study design, data collection and analysis, decision to publish, or preparation of the manuscript. If the funders had no role in any of the above, include this sentence at the end of your statement: "The funders had no role in study design, data collection and analysis, decision to publish, or preparation of the manuscript."

2) DATA REQUEST: Thank you for providing, as a supplemental excel file, the data underlying each figure ("Source_Data_for_Figures"). Please make sure to reference this file in each relevant figure legend. For example, you can add the sentence, "the data underlying this figure can be found in the supplemental file entitled "Source_Data_for_Figures"

3) BLOT AND GEL REPORTING: We require the original, uncropped and minimally adjusted images supporting all blot and gel results reported in an article's figures or Supporting Information files. We will require these files before a manuscript can be accepted so please prepare and upload them now. Please carefully read our guidelines for how to prepare and upload this data: https://journals.plos.org/plosbiology/s/figures#loc-blot-and-gel-reporting-requirements

Please provide the uncropped and minimally adjusted image supporting Figure S1D

**SUBMITTING YOUR REVISION**

*Re-submission Checklist*

*Published Peer Review*

Sincerely,

Luke

Lucas Smith, Ph.D.

Associate Editor

PLOS Biology

lsmith@plos.org

REVIEWS:

Reviewer #1:

The big question that this study set out to investigate is the function of Parafibromin in proliferating cells and how mis-regulation of this gene contributes to tumor formation. In this revised study, Deng et al. made excellent attempt to address the reviewer's critique, and newly disclosed data clarified some questions raised in the previous review. However, these data also suggested that the model put forth in Fig. 7G is overly simplistic and perhaps highly biased. The authors provided plenty of data to suggest that Parafibromin plays a role in regulating asymmetric cell division, cortical cell polarity and the mitotic machinery. Yet, it remains extremely confusing as how Parafibromin regulates cell proliferation. This is largely due to their inability to distinguish direct vs. indirect effect of removing/knocking down Parafibromin function. The reviewer wants to emphasize that this is not due to lack of effort or vigor from the authors themselves, but perhaps, the complexity of biology itself is the reason. The data presented in this study provides a useful framework that others who are interested in Parafibromin can build on in their future work. Thus, the reviewer supports publication of this study after addressing two lines of inquiries outlined below.

1. The reviewer wants to air a deep concern that it is highly unlikely that the plethora of hyx-mutant phenotypes (spindle defects, polarity defects and ectopic neuroblast formation) and the severity of these phenotypes are the consequence of reduced aurA and polo transcription. The discrepancy between nearly complete suppression of ectopic neuroblast formation and minute restoration of cell polarity protein localization is difficult to overlook. This might be partly due to lack of 2 important control genotypes (overexpression of AurA or Polo). The reviewer would like to insist that the authors examine these two control genotypes under identical conditions and integrate the results in the appropriate bar graphs and in their data interpretation. Furthermore, the authors should revise the Y-axis (0-150 instead of 90-150) used in presenting neuroblast quantification in Fig. 7C. As is, the effect of overexpressing AurA or Polo is overly exaggerated.

2. The authors mentioned in the discussion that previous studies suggest a role for Parafibromin in regulating a myriad of cell cycle and cell proliferation genes including CyclinD1 and C-Myc. I would like to see that the authors further elaborate this line of reasoning by incorporating their own data in the discussion. The authors should acknowledge that cell polarity phenotypes observed in hyx-mutant neuroblasts do not resemble either aurA or polo-mutant neuroblasts previously described by the Wang lab. The fact that all known apical and basal proteins simultaneously become cytoplasmically localized in hyx-mutant mitotic neuroblasts strongly suggests that there is collective failure of cell cycle control in multiple aspects. Thus, it is logical to propose that Parafibromin functions as an upstream regulator of numerous cell cycle genes including aurora A, polo, CyclinD1 and C-Myc. The authors should revise Fig. 7G by including this more inclusive model as an alternative to their preferred model or simply replacing their preferred model altogether.

Reviewer #2

The manuscript has greatly improved.

Nevertheless, I still have some comments that need to be addressed before publication (see below).

However, I feel it would be necessary to perform rescue experiments of the Hyx polarity phenotype using active and inactive Polo kinases and not only using the WT kinase (Figure 7). Western Blot against Polo and AurA using Hyx RNAi brains would also be nice.

P2. Abstract: « We have also discovered that Hyx is required for the formation of interphase microtubule-organizing center and mitotic spindles in NSCs.». Spindes do form in Hyx mutants but they are not normal. Please amend.

P5. HyxHT622 is a truncated protein, the anti-Hyx antibody raised against the (1-176 aa) should reveal the 248 aa protein and not the full length protein. FigS1: Why is the truncated HyxHt622 mutant appears at the same size as the WT Hyx protein in Western Blots ? This is a bit suspect.

P8.« Perhaps hyx-depleted cells are small in size and with altered cell fate, thus unable to expand in this tumour assay ». I don't understand why this would be the case. There are examples of small fly brain tumor that exhibits small neuroblast-like cells with altered cell fate and asymmetric cell division defects. These tissues produce lethal tumours when injected in host flies.

P10. The cytokinesis delay or defect description is unclear. The description of the phenotype doesn't make sense. If the author claim there is no cytokinesis defect, the extra Asl dots could be due to overduplication or centrosome fragmentation.

If there is a cytokinesis defect, this should be visible for some of the dividing cells observed by live-microscopy (Figure 2J). I don't believe that speculation on cell fate in this paragraph is needed. It is suggested that Asterless extra dots are dots are due to cytokinesis defect but this is not shown in this paper. These extra dots are not visible in 3J ? 5A in interphase, but visible in 6D and S5 in mitotic. Therefore Centriole fragmentation appears to occur in mitotic cells.

P12.Please perform a Western Blot to state for decreased Polo and AurA protein levels. The proteins might be well expressed but not well located at centrosomes.

P15 « The reported downregulation of gene expression using hyx RNAi likely underestimates the contribution of Hyx to this regulation». I believe this is too speculative and should be kept for discussion or be amended.

P16. « To test the effect of Hyx on cell polarity genes, we performed RT-qPCR on third instar larval brains from hyx RNAi; hyxHT622/+ under the control of insc-Gal4. » Why not using an ubiquitous driver such as Actin-GAL4 or Tubulin-GAL4 instead of Insc that is only expressed in the central brain?

P17 and rebuttal letter : « We have changed the statement to the following sentence including appropriate references on manuscript page 17. The new role for Hyx in regulating microtubule growth and asymmetric divisions in NSCs, proposed in this study is consistent with the previous finding that NSC polarity is dependent on microtubules. "

This is my mistake in the original review: I meant that polarization, as described by Knoblich, Jan, Gonzalez groups, is a MT-independent process. Therefore the effect of Arl2, Msps on polarity is likely MT independent. This is extremely important to revise this statement.

P17. «Therefore, the function of Hyx/Parafibromin in regulating microtubule growth and centrosomal assembly is likely a general paradigm in cell division regulation, which might be disrupted in cancer cells ». There is no effect shown on MT growth in this study, but defective centrosome assembly and centrosomal MT nucleation. Please clarify in the text. Otherwise, it is possible to monitor MT growth using labelled plus end binding proteins

Please start the Y axis at 0 (Fig 7C).

Figure 7G. I was intrigued by the arrow in dotted line between Aurora A and Polo. Does this suggest that Polo is activated by Aurora to promote cell polarisation? I believe this link has been clearly shown in a recent 2022 study. If this is the case please cite the appropriate reference. If Aurora A activates Polo to promote polarization, partial loss of both kinase could be relevant to explain the polarisation defect in Hyx RNAi or mutant cells.

Fig S8G and H. Why gamma Tubulin is still recruited in Hyx RNAi cells although Polo and aurA are strongly decreased on these structures? This is different to what is happening in neuroblasts (Fig S9). Please comment.

Fig S10. Several other Polarity genes are down regulated in addition to AurA and Polo. It is therefore not surprising that AurA and Polo OE cells do moderately rescue Hyx knock down. It should be stated more clearly that Hyx defective polarisation phenotype regulates might be caused by diminished expression of more polarity genes that AurA and Polo.

The rescue effects are very weak and the classification of phenotypes in Figure 7 based on crescent intensity appears highly subjective following AurA and Polo over expression. I believe this could be caused by the fact that overexpressed Polo is not activated by upstream kinases. I am wondering why the active Polo kinase variant (available in flies as UAS-PoloT182D constructs and generated separately by the groups of Archambault and Conde) was not used in the assay. This would have probably led to less ambiguous results.

Reviewer #3: 

Overall, the authors did a nice job addressing the revisions. Of note, they further test their model describing polo and aur A as downstream transcriptional targets of hyx through generation of transcriptional fusions and convincing genetic suppression assays. The revised manuscript also includes improved quantification, additional controls (e.g., ChIP), and expanded statistics and details to the methods, increasing the rigor of the work. Importantly, the authors responded to requests to temper certain conclusions and clarify the molecular framework proposed for Hyx function; specifically, they suggest aspects of PCM recruitment and errant NSC symmetric divisions arise as downstream consequences to down-regulated polo and aur A. Consequently, the manuscript includes increased mechanistic insight and rigor and is now suitable for publication.

---

## [Editor Report · Decision Letter 2]

8 Sep 2022

Dear Dr Wang,

Thank you for your patience while we considered your revised manuscript "Parafibromin/Hyrax Governs Cell Polarity and Centrosome Assembly in Drosophila Neural Stem Cells" for publication as a Research Article at PLOS Biology. I apologize again for our apologies in sending you a decision. This revised version of your manuscript has now been evaluated by the PLOS Biology editors and the Academic Editor. Our Academic Editor has commented that the manuscript is much improved and that it is almost ready for publication, but he/she has a few minor comments that will need to be addressed before publication. I have included these comments, below my signature, and would like to invite one last revision to address these. 

**IMPORTANT: As you address these last issues, we also ask that you attend to the following editorial requests: 

1) Title: We think that the title should be edited slightly for brevity. If you agree, we would suggest changing it to "Parafibromin Governs Cell Polarity and Centrosome Assembly in Drosophila Neural Stem Cells"

2) Data: Thank you for providing the underlying data for each of your figures, as a supplementary file. Please reference this file in each of your figure legends (including supplemental). For example, you can add the statement, "the underlying data for this figure can be found in the supplementary file "Source Data".

3) Data: We require the original, uncropped and minimally adjusted images supporting all blot and gel results reported in an article's figures or Supporting Information files. We will require these files before a manuscript can be accepted so please prepare and upload them now. Please carefully read our guidelines for how to prepare and upload this data: https://journals.plos.org/plosbiology/s/figures#loc-blot-and-gel-reporting-requirements

>>Please provide the uncropped, minimally adjusted images relating to the blots presented in Figures 6P,R and Fig S1D

Based on our Academic Editor's assessment of your revision, we are likely to accept this manuscript for publication, provided you satisfactorily address their remaining points below and the data and other policy-related requests, above.

We expect to receive your revised manuscript within two weeks. 

*Published Peer Review History*

*Press*

Sincerely,

Luke

Lucas Smith, Ph.D.

Associate Editor,

lsmith@plos.org,

PLOS Biology

COMMENTS FROM THE ACADEMIC EDITOR:

Summary:

- “Parafibromin can fully rescue ectopic NSCs and cell polarity defects in Drosophila hyx mutant brains” – although Parafibromin seems to rescue the ectopic NSCs these lineages are not completely wild-type like, this sentence should be changed as it does not fully rescue (even if the authors have not explored the observed lineage defects).

- “Hyx is required for the proper localization of a key centrosomal protein, Polo…” – I think the authors should also mention AurA.

- “overexpression of polo and aurA could significantly suppress ectopic NSC formation and NSC polarity defects” – the significance of these results is not shown in the figures as the authors have not performed a statistical analysis on these data (same in Significance section). Perform statistical analysis.

Results:

- (NSCs/clone) vs. (clones)– uniformize nomenclature.

- “At 10hrs after pupa formation (APF), 88.2% of the NSCs in pupal brains exited the cell cycle…” – NSCs do not synchronously cycle through cell cycle and even in wild-type larval brains the rate of PH3 positive NSCs is much lower than <100%, and has been reported to be anywhere between 20%-50%. Furthermore, in wild type brains it has been shown that the majority of NSCs exit cell cycle ~24h APF. Thus, the percentage of PH3 positive cells does not reflect the number of NSCs that have not yet exited cell cycle, and an analysis at 10hAPF does not allow conclusions about NSC decommissioning. The authors must correct this paragraph and rethink their conclusion. As the time point analysed does not allow any conclusion about NSC cell cycle exit, the authors should include an alternative reason and discuss why they see an increase in PH3 positive NSCs in hyx mutants at 10h APF.

- Fig S2C – missing wild-type Type II clone

- Fig S2C – Rescue by human HRPT2 does not seem to be complete: NB II seem to be smaller than WT NB II (although a control clone is missing); Lineage seems to be disorganized with increased number of Ase+Dpn+ cells and in ectopic locations – this should be discussed and the conclusion about full rescue toned down.

- Analysis of other components of the Paf1 complex – for the RNAis that did not cause a phenotype, the authors should reference manuscripts that confirm that these lines do knock down these genes effectively. The conclusion about the involvement of the Paf1 complex should be adjusted accordingly.

- “knocking down rtf resulted in a weak phenotype type II NSC overgrowth.. (rtf1 RNAi/BDSC#34586: type II, 31.2%, …” – clarify what overgrowth means, i.e. bigger NSCs, more NSCs? What do these percentages mean?

- “…(Fig. 6I; 1.64-fold). Therefore, Hyx promotes the expression of both polo and aurA.” – In this section the authors have not yet shown that Hyx promotes the expression of these genes.

- “By contrast, aPKC mRNA levels were increased to 5.21-fold (Fig S10K)” – This increase is non-significant and therefore this conclusion should be changed. aPKC's high variation should also be discussed.

- Ensure that the Gal4 lines used for each experiment are included in text or in the figure legend (eg. Fig 7A-B)

- Please check definition of all abbreviations (e.g. “BL” or RNAi).

Discussion:

- Although the authors discuss the link between hyx and Polo and AurA they miss to include discussion about the polarity defects observed. I would recommend the authors including a couple of sentences about the polarity defects observed in hyx and also to reference previous literature to support their ideas.

---

## [Editor Report · Decision Letter 3]

16 Sep 2022

Dear Dr Wang,

Thank you for the submission of your revised Research Article "Parafibromin Governs Cell Polarity and Centrosome Assembly in Drosophila Neural Stem Cells" for publication in PLOS Biology. On behalf of my colleagues and the Academic Editor, Catarina Homem, I am pleased to say that we can in principle accept your manuscript for publication, provided you address any remaining formatting and reporting issues. These will be detailed in an email you should receive within 2-3 business days from our colleagues in the journal operations team; no action is required from you until then. Please note that we will not be able to formally accept your manuscript and schedule it for publication until you have completed any requested changes.

PRESS

Sincerely, 

Lucas Smith, Ph.D., Ph.D.

Associate Editor

PLOS Biology

lsmith@plos.org